# Tai Chi increases functional connectivity and decreases chronic fatigue syndrome: A pilot intervention study with machine learning and fMRI analysis

**Kang Wu**[1,2☯], **Yuanyuan Li**[1☯], **Yihuai Zou**[1], **Yi Ren**[1], **Yahui Wang**[1], **Xiaojie Hu**[1], **Yue Wang**[1], **Chen Chen**[1], **Mengxin Lu**[1], **Lingling Xu**[1], **Linlu Wu**[1], **Kuangshi Li**[1]*

**1** Dongzhimen Hospital, Beijing University of Chinese Medicine, Beijing, China, **2** Xinhua Hospital, Tongzhou District, Beijing, China

☯ These authors contributed equally to this work.

* likuangshi89@hotmail.com

## Abstract

### Background

The latest guidance on chronic fatigue syndrome (CFS) recommends exercise therapy. Tai Chi, an exercise method in traditional Chinese medicine, is reportedly helpful for CFS. However, the mechanism remains unclear. The present longitudinal study aimed to detect the influence of Tai Chi on functional brain connectivity in CFS.

### Methods

The study recruited 20 CFS patients and 20 healthy controls to receive eight sessions of Tai Chi exercise over a period of one month. Before the Tai Chi exercise, an abnormal functional brain connectivity for recognizing CFS was generated by a linear support vector model. The prediction ability of the structure was validated with a random forest classification under a permutation test. Then, the functional connections (FCs) of the structure were analyzed in the large-scale brain network after Tai Chi exercise while taking the changes in the Fatigue Scale-14, Pittsburgh Sleep Quality Index (PSQI), and the 36-item short-form health survey (SF-36) as clinical effectiveness evaluation. The registration number is ChiCTR2000032577 in the Chinese Clinical Trial Registry.

### Results

1) The score of the Fatigue Scale-14 decreased significantly in the CFS patients, and the scores of the PSQI and SF-36 changed significantly both in CFS patients and healthy controls. 2) Sixty FCs were considered significant to discriminate CFS ($P = 0.000$, best accuracy 90%), with 80.5% ± 9% average accuracy. 3) The FCs that were majorly related to the left frontoparietal network (FPN) and default mode network (DMN) significantly increased ($P = 0.0032$ and $P = 0.001$) in CFS patients after Tai Chi exercise. 4) The change of FCs in the left FPN and DMN were positively correlated (r = 0.40, $P = 0.012$).

**Data Availability Statement:** All the data and code files are avaulable in github in https://github.com/Clancy-wu/TaiChi-CFS-2022.

**Funding:** The study was supported by funding of the Beijing Natural Fund Committee with number 7204277 and the National Natural Fund Committee with number 82004437. Kuangshi Li is the recipient of these funds. The funders had no role in study design, data collection and analysis, decision to publish, or preparation of the manuscript.

**Competing interests:** The authors have declared that no competing interests exist.

## Conclusion

These results demonstrated that the 60 FCs we found using machine learning could be neural biomarkers to discriminate between CFS patients and healthy controls. Tai Chi exercise may improve CFS patients' fatigue syndrome, sleep quality, and body health statement by strengthening the functional connectivity of the left FPN and DMN under these FCs. The findings promote our understanding of Tai Chi exercise's value in treating CFS.

## Introduction

Chronic fatigue syndrome (CFS), which is also called myalgic encephalomyelitis, is characterized by severe fatigue, muscle weakness, sleep disturbance, disturbances of neuropsychological function, and self-reported impairments in concentration as well as short-term memory [1]. The severe fatigue persists or relapses for six or more consecutive months and cannot be explained by any other medical condition, depressive disorder, alcohol abuse, or severe obesity, as per the clinical evaluation guidance for CFS of the Fukuda criteria [2]. Epidemiological studies indicate that the estimated prevalence of CFS is about 2.5% in the United States [3], 7.8% in Iceland [4], and 0.89% in comprehensive estimation [5]. Generally, the unexplained fatigue and sleep disturbance intensify gradually to overwhelm the patient's daily life, thus deteriorating their life quality and social or familial relationships [6, 7]. There is no known clear mechanism of pathogenesis and effective treatment yet [8, 9]. Since the body is fatigued, whether CFS patients should avoid exercise was a controversial topic in the past [10]. However, with several milepost-type randomized trials demonstrating the usefulness of exercise [11–13] and growing evidence in support of it [14, 15], the view that exercise therapy is a potential effective treatment for CFS has been accepted and recognized [14, 16]. In addition, the National Institute for Health and Care Excellence updated its guidelines to recommend exercise therapy for CFS [17]. Hence, as the value of exercise therapy in treating CFS reaches a consensus, the focus of the discussion on CFS should also include why and where exercise therapy works.

Tai Chi is a kind of historical exercise in Chinese culture, which is composed of some specific kong-fu actions. Generally thinking in Chinese medicine, these kong-fu actions should be completed in a continuous but slow manner, and the exercisers are required to regulate their breath to feel the existence of vital Qi in body. As an important concept of Qigong, Chinese doctors believe that Vital Qi could be induced by Qigong for balancing the internal body situation and external natural state. Moreover, this process can help people to keep health through moderate exercise and meditative breathing holds [18]. Previous clinical trials have concluded that Qigong can improve the symptoms of CFS and its effect is better than general aerobic exercise, because the Qigong can intervene the body and spirit of CFS patients [19, 20]. Therefore, more attentions are deserved on Qigong in the CFS field. As an exercise method with the advantages of Qigong, Tai Chi was reported to have positive effects in CFS [21], sleep quality [22], cancer-related fatigue [23], cognitive impairment improvement [24], and other aspects [25, 26]. Consequently, digging into the mechanism of Tai Chi in the treatment of CFS is of great significance to promote application of exercise therapy in CFS.

Because of the impairment of concentration, sleep disorders, abnormalities in cognition, and a high proportion of psychiatric complications in CFS patients, it is believed that the central nervous system is involved in the process of CFS [27]. Therefore, CFS could be considered a neurological disease [28]. As an advanced method for the study of the human brain, the functional magnetic resonance imaging (fMRI) approach is widely used in neurological diseases.

Previous studies have demonstrated the functional alteration of the brain networks. Boisso-neault [29] revealed that the decreases of the functional connections in several brain networks on CFS patients were related to the fatigue increases. The discovery proved that perturbations of the functional connections may underlie the chronic fatigue. Charles [30] detected the decreased intrinsic connectivity within the left frontoparietal network (LFPN) and the decreased extrinsic connectivity involving the sensory motor network (SMN) and the salience network (SN) on CFS patients, both of which were related to the level of their fatigue symptom. Besides, the default mode network (DMN), which is focused on recently, showed connection disruptions in CFS [29, 31] and other fatigue related disease [32]. Nonetheless, one study declared no significance difference on DMN in CFS patients compared to healthy people [33]. It seemed that multiple brain networks have participated in the causing of CFS, however, the alterations of brain network characteristics in CFS are still required further clarification.

According to previous findings, long-term Tai Chi practice would be benefited for the improvements of the functional connectivity [34, 35] to prevent the cognitive decline, and was superior than general aerobic exercises in eliciting brain plasticity [36]. In addition, a study illustrated the effect of Tai Chi in regulating the rest-state functional connections of the DMN and LFPN to enhance cognitive function on healthy people [37]. Further, others studies indicated that Tai Chi could improve sleep disorders through increasing functional connections of the sub-regions of DMN, including the medial prefrontal cortex and the medial temporal lobe [38, 39]. It seemed that functions of Tai Chi and disfunctions of CFS were overlapped partly in brain networks, for instance, the DMN and LFPN. Consequently, we speculate the reorganization of brain network's functional connectivity via Tai Chi practice would also work for CFS, despite there being little research concentrating on it.

Figuring out the alterations of characteristics and specific brain networks in CFS patients compared to the healthy persons is of significant value. However, it is not easy because of the contradiction in previous studies. The better choice may be machine learning. Machine learning can help identify disease patterns and general rules from large datasets to provide a useful predictive model and novel insights on a disease of interest, and it is being increasingly applied to neurological diseases [40, 41]. Different from traditional statistic methods (e.g., logistic regression), machine learning not only majorly focuses on making predictions and classifications, but also can flexibly process a large but messy dataset without many pre-assumptions [42]. Therefore, combining machine learning and fMRI is a better approach to identify neuro-imaging biomarkers of CFS. To date, three studies exist that have performed machine learning and fMRI to differentiate CFS patients and report potential brain regions of interest [43–45]. Nevertheless, all of these were cross-sectional studies that focused on making predictions. There is no longitudinal study yet, which could help discover knowledge regarding treatment comparisons with respect to specific brain regions.

In this context, the present study designed a longitudinal trial on Tai Chi as the intervention therapy and functional connections (FCs) of whole-brain regions as the exploratory variable, to detect the abnormal brain connectivity structure in CFS and the changing pattern of the structure in response to Tai Chi exercise. We hypothesized that 1) A special pattern in functional brain connectivity may exist to distinguish CFS patients and healthy controls. 2) Tai Chi exercise could improve this special pattern of functional connectivity in large-scale networks to alleviate CFS.

## Materials and methods

### Participants

This trial referred to the previous research paradigm [46]. Twenty CFS patients and 20 volunteers matched in age, gender, and body mass index (BMI) were recruited from 2020-01-

03 to 2021-01-02 at Dongzhimen Hospital affiliated to the Beijing University of Traditional Chinese Medicine. CFS patients were diagnosed by the 1994 Fukuda CDC [2] criteria. They had chronic fatigue that could not be explained by the clinic. In CFS, the chronic fatigue is neither explained by working nor can be relieved by relaxation, and it generally persists or relapses for more than six months and influences the daily work and behavior routines and abilities of patients significantly. The patients in the CFS group and volunteers in the healthy control group (HC) were right-handed with an age range of 25–65 years, no history of mental disorders or psychotropic drug-taking, and no history of Tai Chi practice. Pregnant women, lactating women, girls who had menstruation during fMRI scanning, and severely obese people with BMI more than 45 were all excluded. All subjects submitted informed consent with their handwritten signature, and the Ethics Committee of Dongzhimen Hospital affiliated to the Beijing University of Chinese Medicine provided the ethics committee approval (number DZMEC-KY-2019-195). The registration number of the study is ChiCTR2000032577 in the Chinese Clinical Trial Registry. Forty subjects completed our trial, and no one quit before completion. Fig 1 was the flowchart of the study recruiting.

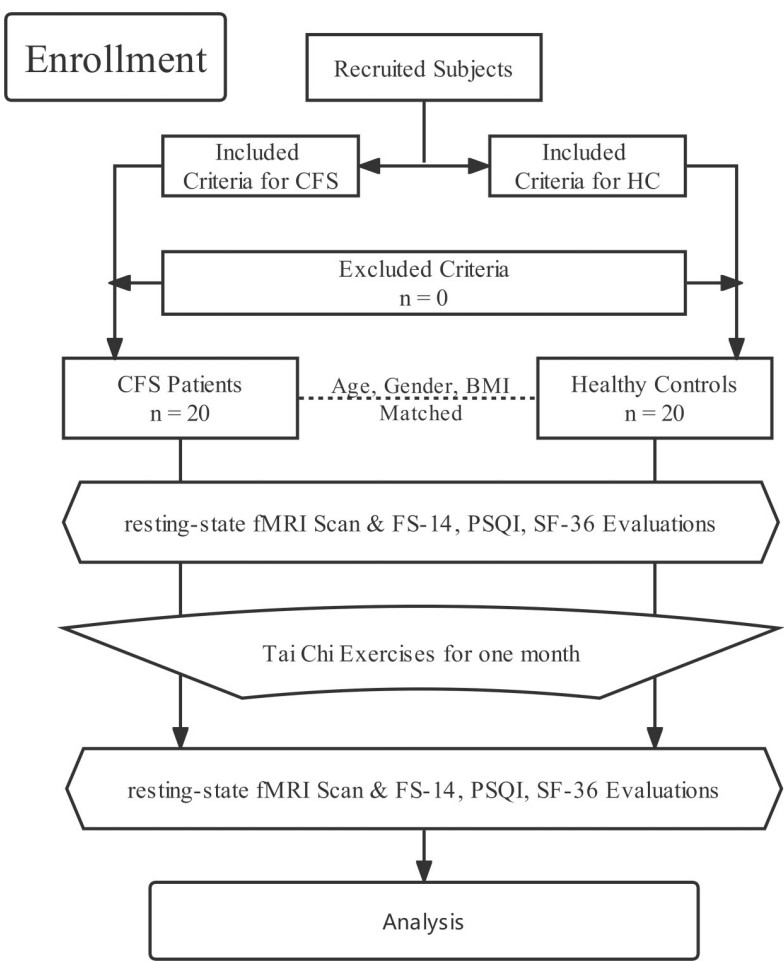

**Fig 1. The flowchart of the study recruiting.**

## Clinical design and evaluation

After subjects entered the group, they received the intervention of Tai Chi for one month and clinical evaluation as well as fMRI scanning twice. The subjects both in the CFS group and HC group performed Tai Chi practice eight times in our hospital, including actions and posture correction, under the professional guidance of three Tai Chi coaches. The Tai Chi in our trial was the simplified 24-style Tai Chi issued by the State Sports Administration in China, and the teaching schedule was two times per week and practice for half hour per teaching session. All coaches in the trial were graduates with sports majors and many years of experience with Tai Chi, and they were required to know the design of the trial and the basic knowledge of CFS disease before they took part in this research. On each teaching class, the subjects whether in the CFS group or HC group were mixed to be taught but coaches never knew about group information. For the rest of the month, subjects were required to practice Tai Chi for 30 minutes per day by themselves at home. During our study, for each subject, the Tai Chi teaching was recorded by live recording, and the family Tai Chi exercise was recorded through video feedback and telephone follow-up, in order to supervise the quality of the practice. In the end, all subjects had completed the required exercise times and exercise frequency.

Before the first Tai Chi exercise and after the last exercise, all subjects underwent clinical evaluation and resting-state fMRI scanning from the scale evaluator, who was blinded about the grouping of subjects in the study. A clinician made each decision about whether subject would be assigned to the CFS or HC group. The clinical evaluation both in two groups was conducted using three scale questionnaires: the Fatigue Scale-14 (FS-14) for fatigue symptom assessment (the higher its score, the more serious is the fatigue); Pittsburgh Sleep Quality Index (PSQI) for sleep quality measurement around one month (the higher its score, the lower is the sleep quality); and the MOS 36-item short-form health survey (SF-36) for people's healthy state evaluation (the higher its score, the healthier is the body).

## fMRI data acquisition

The fMRI data were acquired by a Siemens 3-T MRI scanner (Germany), and the parameters of this machine scan were as follows: the resting-state echo-planar imaging sequence acquisition (time of repetition = 2,000 ms, time of echo = 30 ms, flip angle = 90˚, phase encoding direction = A >> P, coverage = whole brain including cerebellum, field of view = 240 mm × 240 mm, matrix = 64 × 64, slice thickness = 3.5 mm, volumes = 240), the three-dimensional structure imaging adopting T1W1 sequence (time of repetition = 1900 ms, time of echo = 2.53 ms, coverage = whole brain including cerebellum, field of view = 250 mm × 250 mm, matrix = 256 × 256, slice thickness = 1.0 mm, volumes = 176).

## Data pre-processing

The pre-processing of functional and structural images data was performed via the workflow of fMRIprep [47] (version 20.2.1) in DPABISurf [48] (http://rfmri.org/, version V1.6). fMRIprep is an ensemble tool that provides both structural and functional image processes that can automatically generate robust results compared to other tools or software [47]. Briefly, for each subject, the structural image was first skull-stripped and then segmented into cerebrospinal fluid, white matter, and gray matter. Second, the brain surface was reconstructed from the segmented cortical gray matter and registered into the standard MNI152 spatial template through the nonlinear method. The steps of the functional image on volume space involved removing the first 10 time points, voxel size normalizing to 2 mm, slice timing, ICA-AROMA noise removal (a robust method using independent component analysis to remove motion artifacts [49]), head motion correction, MNI152 template spatial normalization, nuisance

covariates regression, smoothing with Gaussian kernel of full width at half maximum of 6 mm, and signal filtering with 0.01–0.1 Hz. All these processes were performed from one subject to another automatically to ensure that the pre-processed result was reliable and repeatable as possible.

## Feature construction

Machine learning is the algorithm to detect potential relationships between target and explanatory variables, these variables are known as the "features". After pre-processing, we used the Schaefer template [50] to extract the time series of each parcellation (usually called the region of interest, ROI) and compute their correlation coefficients in the volume space via DPABI-Surf. This template is built based on the brain network connectome and has 400 ROIs divided into seven large-scale functional brain networks, namely the visual network (VN), somatomotor network (SMN), dorsal attention network (DAN), ventral attention network (VAN), limbic network (LN), frontoparietal network (FPN), and default mode network (DMN). Generally, it is better than the automated anatomical labeling template (AAL template) to discover functional brain connection disorders. The correlation coefficients were then standardized through fisher Z transform (FC value) and combined with the gender, age, BMI, and head motion coefficient. The head motion coefficient was represented by the mean framewise displacement (FD) value of all time points, which was computed from the formulations of Power [51] and Jenkinson [52]. After these procedures, we had 160,004 features and 80 samples (each subject before and after intervention) for machine learning.

## Feature selection and predictive model construction

After the features were constructed, we first selected samples before the intervention and split them into training data and testing data randomly. Considering our small sample size, we set the split ratio to 0.5. Then, after gender was binarized and values were standardized, a linear support vector classification (linear SVC) with L1 norm penalty and regularization parameter (C = 125.9) was run on the training data via scikit-learn [53] (version 1.0.1) to filter the 160,004 features. This linear SVC model assigned each feature an importance coefficient, and the features were then removed if their coefficient was below the default threshold (threshold = 1e-5) under the regularization parameter. Second, we employed random forest (the optimal hyperparameter is three estimators, Gini criterion, two sample split, and one sample leaf) as the predictive model to discriminate CFS and HC with the filtered features. Random forest is an ensemble algorithm and widely used in the study of many neurological diseases [54, 55] because of its high performance and accuracy.

## Predictive model evaluation and feature rank

For our random forest model, we performed five-fold cross-validation in the training data to evaluate the generalization performance and predicted the test data to evaluate the predictive ability. Cross-validation and prediction were both repeated 10 times to compute the average score and standard deviation. For further model assessment, we performed a permutation test 5,000 times to evaluate whether our predictive model was better than dummy classification (a classification that predicts labels per the alternative hypothesis by chance) [56]. In each permutation test, the labels were randomly shuffled for prediction. Our predictive model and the dummy classification were then run to calculate the average score from five-fold cross-validation in the test data. A $P$ value was output by ranking the obtained scores, and a $P$ value $< 0.05$ was considered evidence that the model could significantly distinguish the CFS group and HC group. For feature rank, even though the random forest model would provide each feature an

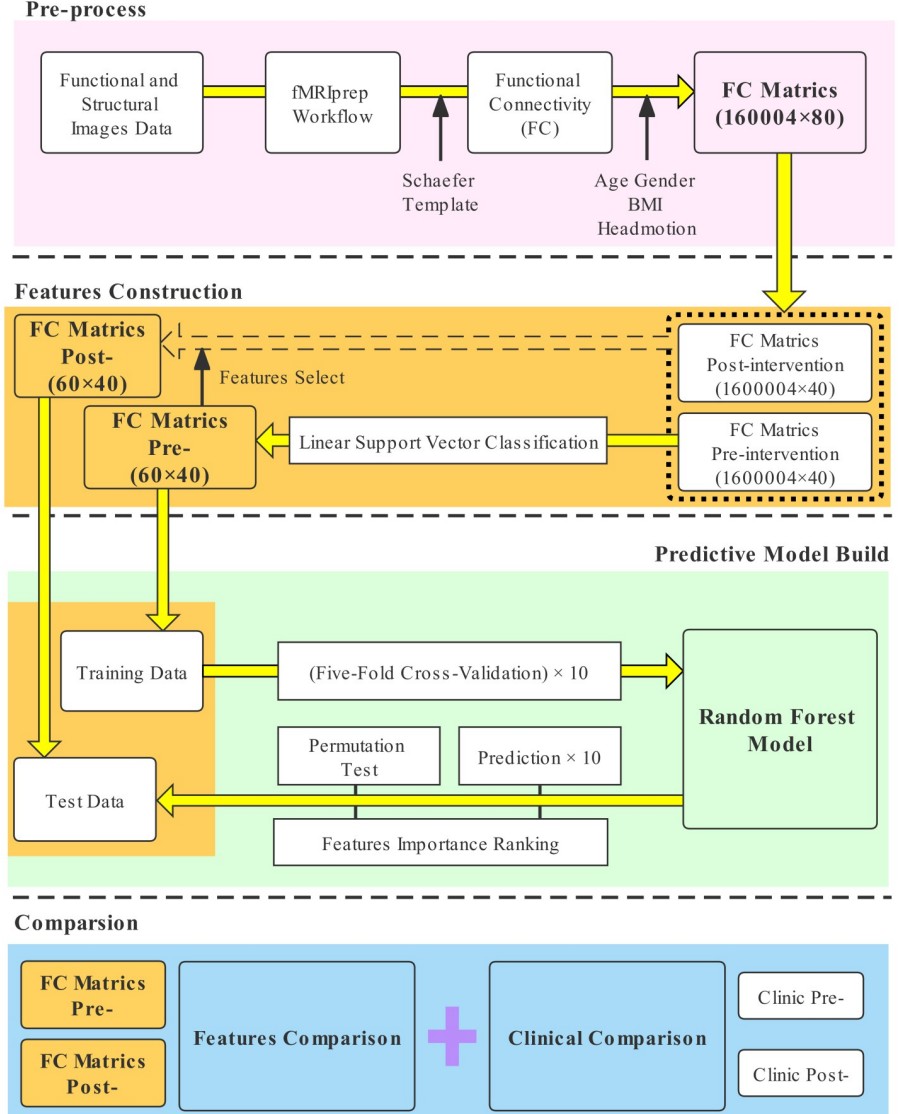

**Fig 2. The workflow of the predictive model building in the study.**

importance coefficient, we still run the permutation test 5,000 times to measure the contribution of the features. This step was also run 10 times to obtain the average feature importance coefficients, which this approach of feature ranking was considered to reduce bias and overcome some potential limitations of a tree-based model [57]. Fig 2 is the workflow of the predictive model building in the study.

## Statistical analysis

All the statistical analyses in this trial were performed in R (version 4.1.2). For demographics, the chi-square test without correction was used for gender comparison; the Wilcoxon test of independent samples was used for age comparison; and Student's $t$-test for independent samples was performed for BMI comparison, questionnaires baseline comparison and questionnaires post-intervention comparison. For clinical evaluation, covariance analysis was

performed within each group and the repeated measures analysis of variance was used for further comparisons, both of which took gender, age and BMI as the confounding factors. Besides, we calculated the average score of the max-min normalization transformation value for the SF-36, which has eight scores to evaluate the body health state along different dimensions. The $P$ value of the three scales were adjusted via Bonferroni method ($\alpha = 0.05 / 3 = 0.016$). For brain networks comparison, due to their Gaussian distribution, the Student's $t$-test for paired samples was used on intra-group comparison, and the covariance analysis for independent samples was used on inter-group comparison. The Bonferroni method for multiple comparison correction was applied (single network: $\alpha = 0.05 / 7 = 0.0071$; bilateral network: $\alpha = 0.05 / 14 = 0.0036$). For the Pearson correlation test, partial covariance analysis was performed for features and clinical measurements with a confounding matrix containing the gender, age, disease course, and BMI. This analysis was also performed between networks with a confounding matrix of values before the intervention. All the above statistics computed the $P$ value in two tails, and $P$ value < 0.05 was considered significant.

## Results

### Demographics and clinical measurement comparison

As shown in Table 1, the CFS group and HC group haven't significant differences in age, gender, and BMI, which suggested that they were comparable in base information.

After the Tai Chi exercise, excluding the influence of confounding factors, the score of FS-14 decreased significantly in the CFS group ($P = 0.000$); the score of the PSQI decreased significantly both in the CFS and HC group (both $P = 0.000$); and the score of SF-36 increased significantly both in the CFS and HC group ($P = 0.001$ and $P = 0.000$).

In Table 2, the FS-14 displayed a significant main effect in group factor (F = 25.658, $P = 0.000$, Partial $\eta^2 = 0.430$) and a non-significant interaction effect in group and time factors; the PSQI showed a significant main effect in group factors (F = 4.836, $P = 0.035$, Partial $\eta2 = 0.125$) and a significant interaction effect in group and time factors (F = 10.556, $P = 0.003$, Partial $\eta2 = 0.237$); the SF-36 showed a significant main effect in group factor (F = 26.807, $P = 0.000$, Partial $\eta2 = 0.441$) and a significant interaction effect in group and time factors (F = 19.721, $P = 0.000$, Partial $\eta2 = 0.367$).

Under a univariate test from the significant interaction effect, the PSQI showed a significant simple effect between CFS and HC groups in pre-intervention (F = 9.302, $P = 0.004$, Partial $\eta2 = 0.215$) and a significant simple effect between interventions in CFS group (F = 24.459, $P = 0.000$, Partial $\eta2 = 0.418$); the SF-36 showed a significant simple effect between CFS and

**Table 1. Demographics and clinical measurements (mean ± SD).**

| Group | CFS | | | HC | | | $P_{\text{between group}}$ | |
|---|---|---|---|---|---|---|---|---|
| N | 20 | | | 20 | | | - | |
| Age | 38.15 ± 12.05 | | | 32.85 ± 12.31 | | | 0.153 | |
| Female | 14 (70%) | | | 13 (65%) | | | 0.736 | |
| BMI | 23.12 ± 2.84 | | | 21.80 ± 3.42 | | | 0.192 | |
| | **pre-** | **post-** | $P_{\text{post-pre}}$ | **pre-** | **post-** | $P_{\text{post-pre}}$ | $P_{\text{pre-pre}}$ | $P_{\text{post-post}}$ |
| FS-14 | 9.60 ± 2.52 | 7 ± 3.23 | 0.000*Δ | 4.45 ± 3.73 | 3.05 ± 3.35 | 0.086 | 0.000*Δ | 0.004*Δ |
| PSQI | 7.10 ± 2.94 | 5.40 ± 3.12 | 0.000*Δ | 4.70 ± 2.25 | 4.55 ± 2.01 | 0.000*Δ | 0.011*Δ | 0.335 |
| SF-36 | 56.91 ± 13.68 | 77.02 ± 11.48 | 0.001*Δ | 82.14 ± 13.04 | 87.60 ± 11.33 | 0.000*Δ | 0.000*Δ | 0.015*Δ |

The * represents $P$ value < 0.05; Δ represents P value < 0.016 for multiple correction with Bonferroni method.

**Table 2. Repeated measures analysis of variance in clinical measurements.**

| | | Repeated Measures Analysis of Variance (Main & Interaction Effect) | | | Univariate Test (Simple Effect) | | | |
|---|---|---|---|---|---|---|---|---|
| | | F | P | Partial Eta Squared | | F | P | Partial Eta Squared |
| FS-14 | Group | 25.658 | 0.000*Δ | 0.430 | | | | |
| | Time | 0.837 | 0.367 | 0.024 | | | | |
| | Group * Time | 0.659 | 0.423 | 0.019 | | | | |
| PSQI | Group | 4.836 | 0.035* | 0.125 | | | | |
| | Time | 0.165 | 0.687 | 0.005 | | | | |
| | Group * Time | 10.556 | 0.003*Δ | 0.237 | Time \| Pre- | 9.302 | 0.004*Δ | 0.215 |
| | | | | | Time \| Post- | 1.349 | 0.254 | 0.038 |
| | | | | | Group \| CFS | 24.459 | 0.000*Δ | 0.418 |
| | | | | | Group \| HC | 0.069 | 0.795 | 0.002 |
| SF-36 | Group | 26.807 | 0.000*Δ | 0.441 | | | | |
| | Time | 2.099 | 0.157 | 0.058 | | | | |
| | Group * Time | 19.721 | 0.000*Δ | 0.367 | Time \| Pre- | 35.119 | 0.000*Δ | 0.508 |
| | | | | | Time \| Post- | 11.210 | 0.002*Δ | 0.248 |
| | | | | | Group \| CFS | 77.880 | 0.000*Δ | 0.696 |
| | | | | | Group \| HC | 5.876 | 0.021* | 0.147 |

The * represents P value < 0.05; Δ represents P value < 0.016 for multiple correction with Bonferroni method; Group includes CFS and HC groups; Time includes pre-intervention and post-intervention.

HC groups in pre-intervention (F = 35.119, P = 0.000, Partial η2 = 0.508) and in post- intervention (F = 11.210, P = 0.002, Partial η2 = 0.248), and also displayed a significant simple effect between interventions in CFS group (F = 77.880, P = 0.000, Partial η2 = 0.696).

## Comparison of model performance and feature selection between interventions

Before the intervention, 160,004 features decreased sharply to 60 after linear SVC filtering (Fig 3A and 3D, S1 Table). With these 60 features, the average score of the cross-validation of our random forest model was 87% ± 7% and the prediction score was 80.5% ± 9% (S4 Table). Fig 3B shows the result of the permutation test applied 5,000 times, and it can be seen that our model could distinguish CFS and HC significantly (P = 0.001, accuracy = 90%), whereas the dummy classification could not (P = 1, accuracy = 45%). After the intervention, the random forest model's prediction score decreased to 53.5% ± 3.9%, with no significance in the permutation test (P = 0.675, accuracy = 47.5%, S1 Fig). The result of the feature importance ranking in the permutation test showed that their average score ranged from −0.1 to 0.1, and the maximum feature importance coefficient was ROI 368−238, whose average score was 0.2 ± 0.09 (Fig 3C, S2 Table). Comparing before and after intervention, the ROI 363−238 increased significantly in the CFS group (P = 0.016).

## Comparison of brain networks between interventions

In large-scale brain networks, the 60 features that start at VN or end at VN or both were 10%; the SMN was 21.67%; the DAN was 25%; the VAN was 40%; the LN was 20%; the FPN was 28.33%; and the DMN was 45%. Before the intervention, the CFS group and HC group were significantly different in the features of VN, DAN, FPN, and DMN (P < 0.0071). After the intervention, the DMN was significantly increased in the CFS group (P = 0.002), and none of

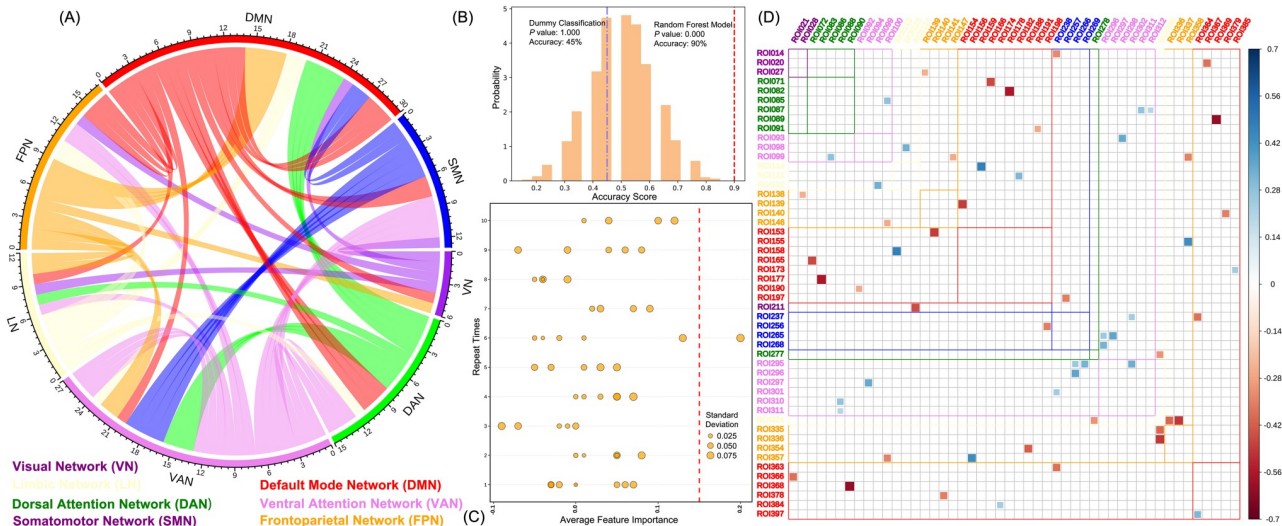

**Fig 3. Model performance and feature selection.** (A) Flow diagram of 60 features filtered by the linear support vector classification. The different colors represent different networks. (B) The result of 5,000 times permutation test in model performance evaluation. The red line is the result of the random forest model, and the blue line is the dummy classification. (C) The result of 5,000 times permutation test, which was repeated 10 times, in feature importance ranking. The size of the circle represents the standard deviation score, and the red line is the importance coefficient = 0.15. (D) The correlation diagram of 60 features filtered by the linear support vector classification. The different colors in ROI represent different networks.

the brain networks showed statistical differences between the CFS group and the HC group (Table 3).

When considering the bilateral nature of the brain network, before the intervention, the left DAN, right LN, bilateral FPN, and bilateral DMN were significantly different in the CFS group and HC group ($P < 0.0036$). After the intervention, the left FPN was significantly increased in the CFS group; thus, the CFS group and HC group lost their difference in brain networks (Table 4). Between interventions, 20 features changed in the CFS group and two changed in the HC group (Figs 4 and 5, S3 Table) but both without significance after correction.

## Partial covariance analysis of features and clinical measurements

Relationships between the DMN and left FPN with clinical measurements were detected in the CFS group. Before the intervention, the FC values of ROI 363–238 had a significant moderate-

**Table 3. Network comparison of features (mean ± SD).**

| Network | Percentage | CFS | | | HC | | | $P_{\text{between group}}$ | |
|---|---|---|---|---|---|---|---|---|---|
| | | pre- | post- | $P_{\text{post-pre}}$ | pre- | post- | $P_{\text{post-pre}}$ | pre- | post- |
| VN | 10% (6) | −0.65 ± 0.56 | −0.30 ± 0.63 | 0.040* | 0.06 ± 0.84 | −0.16 ± 0.63 | 0.324 | 0.002*Δ | 0.485 |
| SMN | 21.67% (13) | 0.90 ± 1.03 | 0.80 ± 0.98 | 0.772 | 0.65 ± 0.76 | 0.41 ± 0.85 | 0.365 | 0.411 | 0.178 |
| DAN | 25% (15) | −0.73 ± 0.72 | −0.10 ± 1.08 | 0.028* | 0.39 ± 1.02 | −0.27 ± 0.89 | 0.014* | 0.001*Δ | 0.575 |
| VAN | 40% (24) | 3.33 ± 1.74 | 2.44 ± 1.92 | 0.153 | 1.81 ± 1.81 | 1.50 ± 1.2 | 0.534 | 0.012* | 0.077 |
| LN | 20% (12) | −0.10 ± 0.78 | −0.11 ± 0.76 | 0.977 | 0.30 ± 0.93 | 0.09 ± 0.83 | 0.412 | 0.091 | 0.448 |
| FPN | 28.33% (17) | −0.36 ± 1.23 | 0.32 ± 1.07 | 0.019* | 1.44 ± 1.38 | 0.80 ± 1.53 | 0.096 | 0.000*Δ | 0.268 |
| DMN | 45% (27) | −1.01 ± 0.82 | −0.07 ± 0.79 | 0.001*Δ | 1.03 ± 1.40 | −0.01 ± 1.69 | 0.072 | 0.000*Δ | 0.888 |

The * represents $P$ value < 0.05; Δ represents $P$ value < 0.0071 for multiple correction with Bonferroni method.

**Table 4. Bilateral network comparison of features (mean ± SD).**

| Network | | Number | CFS | | | HC | | | $P_{between\ group}$ | |
|---|---|---|---|---|---|---|---|---|---|---|
| | | | pre- | post- | $P_{post-pre}$ | pre- | post- | $P_{post-pre}$ | pre- | post- |
| VN | L | 5 | −0.54 ± 0.56 | −0.27 ± 0.57 | 0.0869 | 0.00 ± 0.72 | −0.13 ± 0.63 | 0.4326 | 0.009* | 0.455 |
| | R | 1 | −0.11 ± 0.13 | −0.03 ± 0.16 | 0.0961 | 0.06 ± 0.20 | −0.03 ± 0.20 | 0.2642 | 0.005* | 0.988 |
| SMN | L | 0 | - | - | - | - | - | - | - | - |
| | R | 13 | 0.90 ± 1.03 | 0.80 ± 0.98 | 0.7717 | 0.65 ± 0.76 | 0.41 ± 0.85 | 0.3648 | 0.411 | 0.178 |
| DAN | L | 13 | −0.66 ± 0.67 | −0.17 ± 1.09 | 0.0767 | 0.23 ± 0.92 | −0.30 ± 0.76 | 0.0212* | 0.002*Δ | 0.635 |
| | R | 2 | −0.06 ± 0.29 | 0.07 ± 0.24 | 0.1221 | 0.16 ± 0.42 | 0.04 ± 0.26 | 0.2766 | 0.031* | 0.694 |
| VAN | L | 11 | 1.29 ± 0.82 | 1.22 ± 1.06 | 0.8056 | 1.03 ± 0.95 | 0.71 ± 0.83 | 0.1858 | 0.381 | 0.095 |
| | R | 15 | 3.15 ± 1.71 | 2.15 ± 1.63 | 0.0678 | 1.48 ± 1.79 | 1.49 ± 0.91 | 0.9857 | 0.005* | 0.122 |
| LN | L | 6 | 0.20 ± 0.59 | −0.12 ± 0.47 | 0.0502 | −0.24 ± 0.40 | −0.18 ± 0.64 | 0.7325 | 0.013* | 0.762 |
| | R | 6 | −0.30 ± 0.66 | 0.01 ± 0.67 | 0.1334 | 0.54 ± 0.93 | 0.27 ± 0.45 | 0.177 | 0.000*Δ | 0.176 |
| FPN | L | 8 | 0.02 ± 0.74 | 0.57 ± 0.81 | 0.0032*Δ | 1.05 ± 0.90 | 0.68 ± 1.19 | 0.1533 | 0.000*Δ | 0.738 |
| | R | 9 | −0.38 ± 0.70 | −0.26 ± 0.67 | 0.5541 | 0.39 ± 0.78 | 0.11 ± 0.53 | 0.1341 | 0.002*Δ | 0.063 |
| DMN | L | 18 | −0.41 ± 0.72 | −0.03 ± 0.59 | 0.1182 | 0.60 ± 0.93 | 0.05 ± 1.39 | 0.2008 | 0.000*Δ | 0.791 |
| | R | 11 | 0.22 ± 0.65 | 0.41 ± 0.55 | 0.3609 | 1.06 ± 0.72 | 0.45 ± 1.03 | 0.0344* | 0.000*Δ | 0.898 |

The L represents left and R represents the right of the brain network.

* represents P value < 0.05, and

Δ represents P value < 0.0036 for multiple correction with Bonferroni method.

intensity positive correlation with the PSQI (r = 0.52, *P* = 0.039) after correcting for the influences of gender, age, disease course, and BMI, whereas the values of the DMN and left FPN did not show significant correlation with any clinical measurements or demographic characteristics. However, for the difference between post- and pre-intervention, the changes of values of the left FPN and SF-36 displayed a moderate-intensity negative correlation (r = −0.55, *P* = 0.028, S2 Fig) but without significance under Bonferroni correction. The changes of values of the left FPN and DMN showed a significantly positive correlation (r = 0.40, *P* = 0.012, Fig 6, S3 Fig).

## Discussion

This study identified 60 important FCs that could contribute to discriminating between CFS patients and healthy volunteers via a machine learning approach and a series of robust fMRI methods. From the changes of these FCs between pre- and post- intervention in this longitudinal trial, the results demonstrated that Tai Chi exercise could increase the FCs of the default mode network and the left frontoparietal network to alleviate CFS and improve sleep quality as well as body health state in CFS patients. These findings strengthen our understanding of the mechanism of Tai Chi and provide neural image evidence for Tai Chi exercise treatment for CFS.

Previous machine learning literature regarding CFS reported predictive models with average accuracy near 79% [43–45] and the best accuracy of 82% [43]. Our random forest model had an average accuracy of 80.5% ± 9%, and the best accuracy was near 90%. Besides, we obtained a significant *P* value of 0.001 in the permutation test (Fig 3B), which shows that our model performance is better. This means that the 60 FCs filtered by linear SVC may contain the potential real importance structure, which could help to identify neural biomarkers for discriminating CFS patients and healthy individuals (Fig 3A and 3D). Nevertheless, when we tried to rank these features, none of which was found to play a decisive role (Fig 3C) in our

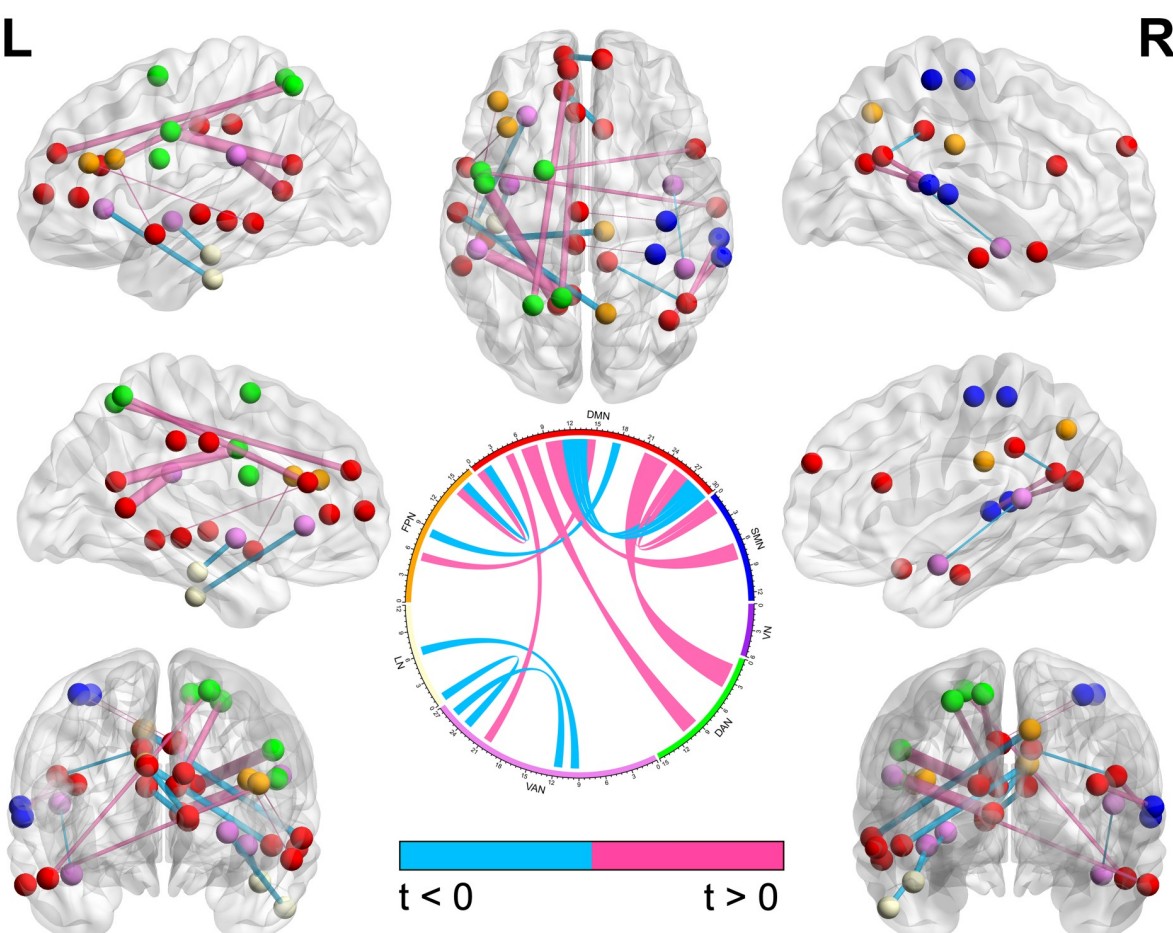

**Fig 4. The changed functional connections of brain nodes in the CFS group with $P < 0.05$ between interventions.** The different colors of brain nodes represent different brain networks. The thickness of nodes' connections represents the statistical value between interventions; the statistical value that was decreased after the intervention is displayed with blue color and that which was increased with pink color.

analysis and even did not appear in repeat instances of the permutation test. This indicates that predicting CFS disease is a comprehensive evaluation process that requires multiple FCs, while the ROI 363–238 in the Schaefer template is just relatively more important among them.

Regarding large-scale brain networks, our results showed that most of these 60 important FCs belong to two different brain networks (90%, Fig 3A), of which the DMN had the greatest relation (45%, Table 3). Similar to our result, various networks that participate in the process of CFS have been mentioned in previous studies, such as the DMN [31, 33, 58], salience network [30, 59], FPN [30], and SMN [30], while the DMN is the most widely reported in recent years. DMN is a classic brain network in the brain's intrinsic activity, involving in memory retrieval [60] and mind-wandering [61]. In current understanding, it generally begins from a high baseline activity and always reduces during attention-demanding tasks [62]. Previous literature have found a significantly lower functional connectivity in the DMN of CFS patients compared with healthy controls [31, 33, 58]. These authors proposed a hypothesis to explain this connection absence, namely that the lower connectivity in the DMN requires more energy support from the brain, which leads to the reduction in other activities, thus aggravating the sense of fatigue [31, 63]. Our study found a similar connection disruption in the DMN. In

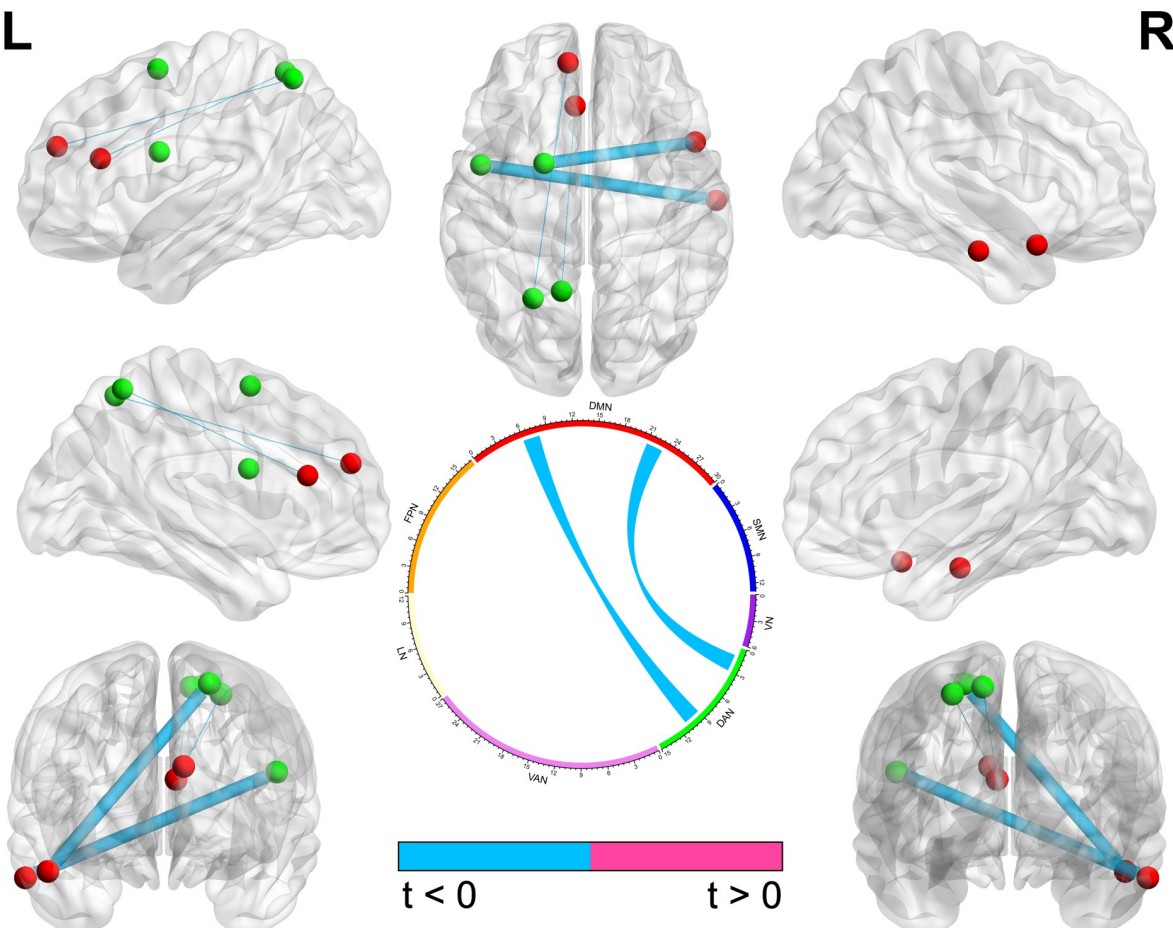

**Fig 5. The changed functional connections of brain nodes in the HC group with *P* < 0.05 between interventions.** The different colors of brain nodes represent different brain networks. The thickness of nodes' connections represents the statistical value between interventions. The statistical value that was decreased after the intervention is displayed with blue color and that which was increased with pink color.

Table 3, the DMN-related FCs in the CFS group are significantly lower than the HC group, and they are significantly increased after the Tai Chi exercise. Based on the above hypothesis, we rationally speculate that long-term Tai Chi practice may enhance the extrinsic connection of DMN to reduce the burden of brain activity consumption, and lessen the fatigue symptom. However, the increased FCs of the DMN in our study were inter-network rather than intra-network. As shown in the previous literatures, most scholars investigated the changes within the DMN of CFS subjects and found the connections disruption of DMN was associated to the fatigue [31, 32, 64]. Besides, the sub-regions connections absence in DMN may also be related to the sleep disorder [38, 39]. Differently, we found that the increases of functional connections between the DMN and other networks could be the reasons of improvements of fatigue and sleep disorder (Tables 1–4). For this result, we infer that there are two possibilities that can be considered. The first possibility is that the defects of machine learning led to the omission of some subsystems of DMN in the calculation, but the changes of these subsystems may alleviate the fatigue of patients. The second possibility is that the increases of functional connections between DMN and other networks may imply an undetected potential mechanism which may be related to the improvement of clinical symptoms in patients with CFS. This mechanism will provide a new direction for the brain network research of CFS in the future.

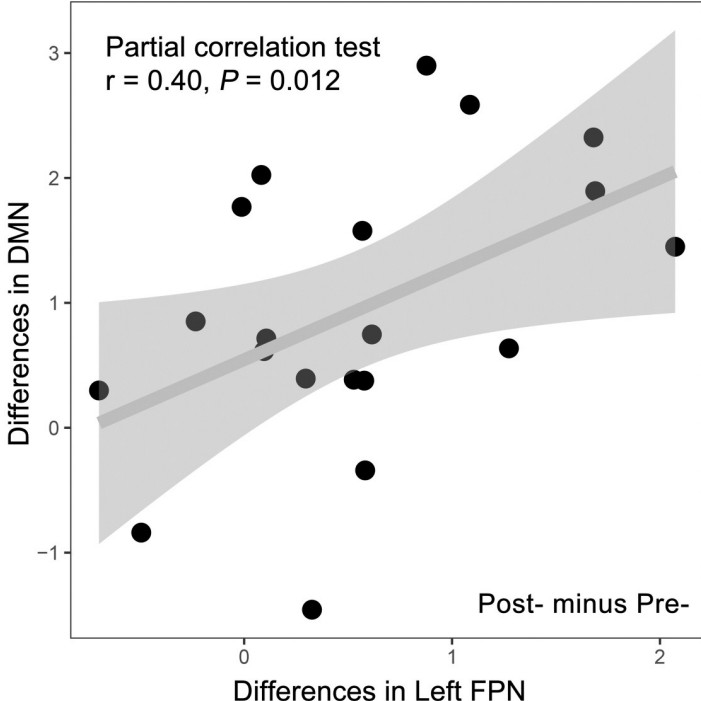

**Fig 6. The result of the partial correlation test in the CFS patient group.** The result of the partial correlation test between changes of the left frontoparietal network and the default mode network in the CFS group. All the changes and differences were calculated by post- minus pre-.

Another notable network is the left FPN. The FPN is considered as a coordinating bridge for diverse tasks according to internal or external demands [65] performed through multi-network activity, and this bridge is mainly used for the balance of the DMN and DAN [66] to participate in the process of cognitive control [67]. A meta-analysis demonstrated that the FPN was the responsible network for the role of regular physical exercise in preventing cognitive decline, particularly on the left side [68]. In Table 4, our study found a significantly lower value of the left FPN in the CFS group compared with the HC group before the intervention, which has been reported before [30] but seldom paid attention to since. After the Tai Chi exercise, the left FPN significantly increased and had no difference with the HC group. More importantly, the increase of the left FPN and DMN exhibited a significant correlation after eliminating the effect of values in pre-intervention (r = 0.40, *P* = 0.012, Fig 6). The DMN and left FPN are a pair of brain network coupling that associated with the cognitive function [69], which often present in a low-level manner in patients with cognitive impairment [70]. Gao-Xia Wei's research indicated that long-term Tai Chi practice could establish a body feedback road to moderate cognitive control system via strengthening the connections between DMN and left FPN, and enhanced the cognitive control capacity to facilitate mental and body health statement [37]. Interestingly, we got similar results in our SF-36 scores (Table 1), and our trial further showed a higher effect size for the health statement of Tai Chi on CFS group than HC group (Table 2). We suppose this interaction of DMN and left FPN may not only increase cognitive control ability, but also optimize connections of DMN to further decrease brain energy cost. This could be a potentially mechanism of Tai Chi practice alleviating CFS. However, whether the increase of the left FPN and DMN occurred simultaneously or successively remains to be explored.

In the feature importance ranking, the FC of ROI 363–238 (starting at the DMN and ending at the SMN) was found to be much important than the others (Fig 3C), and it had moderate intensity positive correlation with the PSQI. However, we recommend that all the FCs we found should be considered important and also the several FCs that were not found in this trial, because machine learning only plays role in prediction rather than statistical inference [42]. Therefore, focusing on the alterations in the whole pattern of the FCs is much more meaningful than analysis of a single FC. After the Tai Chi exercise, owing to the disappeared differences of the VN, DAN, FPN, and DMN between the CFS and HC group (Table 3), our predictive model lost the ability to recognize CFS patients (S1 Fig). This could be supplementary evidence for the effectiveness of Tai Chi in treating CFS. Nevertheless, due to the lack of a follow-up investigation, how long this disappearance persists can be a topic of the future research. Finally, the comparative result of clinical evaluations showed that Tai Chi exercise moderated the fatigue syndrome and improved the sleep quality as well as body healthy scores of CFS patients, which illustrates the effectiveness of Tai Chi exercise for CFS alleviation (Tables 1 and 2). However, since the SF-36 score of CFS patients we included was 56.91 on average, it should be stated that the effectiveness of Tai Chi in our trial is suitable for patients in the mild and moderate sub-healthy category, and more serious symptoms need to be explained by follow-up studies.

The limitations of our study are in five aspects. First, the sample size of this trial was not large enough, and the conclusions we draw require verification through further experiments. Second, due to the randomness of the machine learning algorithm we chose, we may not have found all the important functional connections, which means our conclusions require further research to verify. Third, the PSQI and SF-36 are self-reported questionnaires that describe the health status from a month ago to the present, which means our results had not displayed the clinical efficacy size of Tai Chi exercise fully. Forth, although the denoise movement artifacts were removed, the other artifacts such as respiratory, cardiac and MRI device-based noise might remain influences to our data analysis. Lastly, a waitlist-control group was lacking, and our effectiveness evaluation could be overestimated.

## Conclusions

The present study discovered 60 important FCs for CFS patients via a machine learning algorithm and functional magnetic resonance imaging. The changes in these important FCs demonstrated that Tai Chi could strengthen functional connections of the left FPN and DMN to improve fatigue symptoms, sleep quality, and body healthy statement. Further, the changes in the left FPN and DMN were positively correlated. These findings promote our understanding of the mechanism of Tai Chi in treating CFS.

## Supporting information

**S1 Table. The 60 features after features selected by linear SVC model.**
(PDF)

**S2 Table. Features importance permutation test result.**
(PDF)

**S3 Table. Changes on features after the intervention.**
(PDF)

**S4 Table. The score of the prediction accuracy in each repeat time.**
(PDF)

**S1 Fig. Permutation test of the random forest model in the post-intervention data.**
(TIF)

**S2 Fig. The changes on scores of the SF-36 in CFS patients.** The colors of circle represent different patients.
(TIF)

**S3 Fig. The changes on functional connection values of left FPN in CFS patients.** The colors of circle represent different patients.
(TIF)

**S4 Fig. The ROC score of the random forest model.**
(TIF)

**S1 File. A supplementary statistical analysis based on the linear mixed effects model.**
(PDF)

## Acknowledgments

We appreciated all the subjects that finished our trial. A special thanks should be sent to the colleagues in Xinhua Hospital, that they gave me much care in my trough time. Further, we thank LetPub for its linguistic assistance during the preparation of this manuscript.

## Author Contributions

**Conceptualization:** Kuangshi Li.

**Data curation:** Kang Wu, Yi Ren, Yahui Wang, Xiaojie Hu, Yue Wang, Chen Chen, Mengxin Lu, Lingling Xu, Linlu Wu.

**Formal analysis:** Kang Wu.

**Funding acquisition:** Kuangshi Li.

**Investigation:** Yihuai Zou, Kuangshi Li.

**Methodology:** Kang Wu, Kuangshi Li.

**Resources:** Kang Wu, Yuanyuan Li, Yi Ren, Yahui Wang, Xiaojie Hu, Yue Wang, Chen Chen, Mengxin Lu, Lingling Xu, Linlu Wu.

**Software:** Kang Wu.

**Supervision:** Yihuai Zou, Kuangshi Li.

**Validation:** Kang Wu, Yuanyuan Li, Yi Ren, Kuangshi Li.

**Visualization:** Kang Wu.

**Writing – original draft:** Kang Wu, Yuanyuan Li, Kuangshi Li.

**Writing – review & editing:** Kang Wu, Yuanyuan Li, Kuangshi Li.

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
