## [Decision Letter · Decision Letter 0]

7 Jul 2022

PONE-D-22-14151Tai Chi Increases Functional Connectivity and Decreases Chronic Fatigue Syndrome: A Pilot Intervention Study with Machine Learning and fMRI AnalysisPLOS ONE

Dear Dr. Li,

Thank you for submitting your manuscript to PLOS ONE. After careful consideration, we feel that it has merit but does not fully meet PLOS ONE’s publication criteria as it currently stands. Reviewer's indicate that the manuscript needs some substantial changes. Therefore, we invite you to submit a revised version of the manuscript that addresses the points raised during the review process.

We look forward to receiving your revised manuscript.

Kind regards,

Burak Yulug

Academic Editor

PLOS ONE

Journal Requirements:

"The study was supported by funding of the Beijing Natural Fund Committee with number 7204277 and the National Natural Fund Committee with number 82004437.

The funder Kuangshi Li is the correspondence of this manuscript."

Reviewers' comments:

Reviewer's Responses to Questions

**Comments to the Author**

1. Is the manuscript technically sound, and do the data support the conclusions?

Reviewer #1: Yes

Reviewer #2: Yes

Reviewer #3: Partly

2. Has the statistical analysis been performed appropriately and rigorously? 

Reviewer #1: Yes

Reviewer #2: Yes

Reviewer #3: Yes

3. Have the authors made all data underlying the findings in their manuscript fully available?

Reviewer #1: No

Reviewer #2: No

Reviewer #3: No

4. Is the manuscript presented in an intelligible fashion and written in standard English?

Reviewer #1: Yes

Reviewer #2: Yes

Reviewer #3: Yes

5. Review Comments to the Author

Reviewer #1: The statistical analysis is somewhat simplistic, but serves the purpose. An alternative, potentially more coherent analysis would have been to use a linear mixed effects model framework to assess the interaction of treatment and time, and to evaluate the various contrasts of interest (potentially with more power).

Can the authors remark on whether the functional connections (FCs) that the authors found can be defined in a new subject? Or, are these only able to be retrospectively defined? How would they be measured in a new subject?

With regard to the limitations of the study, given the amount of data collected, is it possible that even an equally-sized set of completely random data would yield a similar number of FCs? This is an issue in large-scale genomics studies.

Line 53: This statement about the relationship of Tai Chi to the broader area of Qigong is not necessarily universally held. It may be better to say that in some lineages of Tai Chi and with some teachers, there can be substantial overlap with practices found in Qigong. I note that in many cases, taijiquan is taught with a completely different emphasis and with completely different defining characteristics, especially if the goal is martial usage of jin and qi.

Line 60: Again, it is quite debatable whether Tai Chi is a representative Qigong. This also ignores the issue of whether the state defined simplified 24-style is an adequate representation of taijiquan in general. And, obviously, the teacher has the primary effect in terms of whether a student can actually learn the skills required for qigong or taijiquan.

Line 77: Change to "could be a mechanism".

Line 85: This absolute statement about "traditional statistical [sic] methods" is quite overblown. In fact, many statistical techniques make relatively few assumptions or relatively mild assumptions. It can just as easily be said that most machine learning techniques suffer from a complete lack of theoretical underpinning, interpretability, or generalizability.

Line 98: What would be the mechanism of action for improving the special functional connectivity?

Line 130: The phrase "was unclear" implies partial knowledge even though full knowledge was expected. Do the authors mean "blinded" here?

Line 204: Why was the max-min normalization performed? It seems unnecessary given the statistical methods used.

Line 313: "[M]achine learning is better at prediction than statistical inference" is likely a truism because machine learning pays nearly no attention to the issues of statistical inference.

Line 347: Does the phrase "at the bottom of my life" correctly interpret the author's intent?

Figure 2: Change "Liner" to "Linear". Also, clarify that feature construction was performed only on training data.

Reviewer #2: Material methods

Line 102: There is no power analysis, to define sample size. I would like to see a power analysis to be sure whether the sample size is enough or not for the fMRI analysis. Why do you choose 20 patients and 20 healthy control?

Line 138: MRI recording parameters have been given really weak. The structural (T1w) and functional recording parameters should be more clear and more details in terms of repeatability.

Line 151: ICA-AROMA tool is only capable of denoise movement artifacts. But the other artifacts such as respiratory, cardiac, or MRI device-based noise remain intact in resting-state data. Did the authors remove artifacts also manually? If they did not it should be written as a limitation.

Authors use linear SVC for feature ranking, and then the most informative features are used to train a Random Forest classifier. The authors must justify why they did not perform classification with SVC. Besides, the feature ranking by SVC is based on having maximum accuracy by the SVC classifier. Can we guarantee that the same features will give the maximum classification result by the Random Forest? Since they already started with SVC we would like to see the classification results with SVC.

It would be better if we can see a more detailed comparative analysis of feature ranking and classification processes. For example (Line 224), they selected 60 of more than 160000 features. How the threshold value of 60 was determined? It would be great if we can see a plot of a number of most informative features vs obtained classification accuracy.

Line 186: Since the dataset is small, the cross-validation is wise to apply, but can we guarantee that 10 repeated runs would be theoretically enough to see the actual performance? A mathematical justification of this value “10” would be required. A plot of average accuracy vs. the number of repetitions would be helpful as well.

Reviewer #3: The authors present work showing that resting state functional network connectivity differs between patients with chronic fatigue syndrome and healthy volunteers, and that a one-month Tai Chi intervention leads to changes in some of these functional networks including the default mode network.

The main hypotheses of this paper are very general. The introduction cites some studies about the effects of Tai Chi in general, but the authors just say that machine learning will find some “special pattern” of FC differences in CFS, and that the regions/networks/connections showing differences at baseline will show some kind of non-specific changes after the training. While the authors discuss some of the particular changes in the context of the literature in their Discussion, the Introduction would be improved by at least including some more specific background information about which functional networks may be affected in CFS, and why Tai Chi may change those networks.

In the “Clinical design and evaluation” section, the authors state that a clinical evaluator was “unclear about the grouping of subjects in the study.” Does this mean they were blinded? Or something else? If there was no waitlist or other control group and everyone received the same Tai Chi intervention, what were the evaluators unclear about?

The authors state that “subjects were required to practice Tai Chi for 30 minutes per day by themselves at home.” Do the authors have data to report about adherence to this requirement (i.e., how much home practice was completed by each participant)? It then says “the Tai Chi teaching and family Tai Chi exercise of each subject were recorded, including live recording…” Does this mean that each home practice session was recorded in some way as well, or just the group or instructor-led sessions? Please clarify.

The data acquisition parameters require more details and clarification. No anatomical scan is mentioned, only stating the “parameters of this machine scan.” The parameters reported appear to be for a functional EPI sequence, but this is not specified. Also, while the time of repetition is stated as 2mm, the total number of volumes (and the total duration of the scan) is not reported.

The data preprocessing section states that smoothing was done at 2mm FWHM. This is much lower than us conventionally used. Typical guidelines suggest to use a FWHM equivalent to 2 or 3 times the size of the voxels. With a voxel size of 3.75x3.75x5mm as stated in the data acquisition section, we would expect a spatial smoothing kernel of at least 7-8mm FWHM. This step can have a significant impact on functional connectivity findings. Can the authors explain this unconventional preprocessing choice?

In the Discussion, the words “decrease” and “lower” are used interchangeably, leading to some confusion as to whether it is referring to the assessment of baseline differences between CFS and healthy volunteers or to the changes resulting from the intervention. For instance, the sentence “Our trial found a similar decrease in the DMN” is confusing. The “trial” should refer to the Tai Chi intervention, but earlier the authors state that increased DMN FC was found following the intervention. So when discussing these results, the authors should clarify the distinction between the baseline differences and the changes observed following the trial/intervention.

The discussion could be improved with more interpretation of the results related to the DMN (i.e. the post-intervention increase in DMN connectivity). The authors currently say that DMN is important and “plays a central role in healthy people and patients with various diseases”, but do not elaborate much. They mention a previous hypothesis of DMN underconnectivity in CFS patients, which suggests that DMN underconnectivity is associated with greater energy needs and reduced energy available to the individual. They say that Tai Chi could “improve the abnormal pattern of the DMN,” but they do not elaborate on this point. This seems like potentially the most important part of the discussion, so it would be good to see this result discussed more. They go on to highlight that increased inter-network DMN connectivity was found after the intervention, and state that this is “interesting”. This could be improved by a couple sentences/references explaining why this might be interesting and worthy of further investigation. The discussion of FPN-DMN connectivity that follows is great!

The axis labels for Figure 6A and 6B say “differences” but this could be clarified by stating whether this is post-pre or pre-post. What are the units of these differences?

---

## [Author Response · Author response to Decision Letter 0]

16 Aug 2022

Response to Review 1

Comment 1: The statistical analysis is somewhat simplistic, but serves the purpose. An alternative, potentially more coherent analysis would have been to use a linear mixed effects model framework to assess the interaction of treatment and time, and to evaluate the various contrasts of interest (potentially with more power).

Response: Thanks for your kind reminders. We used the linear mixed effects model analysis, and its results can be found in the supplementary file (S1 File in Supporting Information). 

As you expected, the interaction effect of the treatment and time was significant (P = 0.002). However, on the pairwise comparisons of simple effect analysis, the FC changes between interventions in CFS group (P = 0.028) and healthy group (P = 0.016) were significant in the single large-scale network (α = 0.05) level, but not the whole network (α = 0.05/7 = 0.007) level. Thus, the linear mixed effects model analysis was not selected. 

To avoid the random errors in the individual level, we performed the covariance analysis on inter-group comparison, taking gender, age, BMI and head motion as the confound matrix, instead of the independent student t test in before. As were showed in Table 2 and Table 3, the results of covariance analysis were very similar to before. [Line 272, Line 280, Pg 12-13]. 

Comment 2: Can the authors remark on whether the functional connections (FCs) that the authors found can be defined in a new subject? Or, are these only able to be retrospectively defined? How would they be measured in a new subject?

Response: Thanks for your kind reminders. The FCs can be found in every subject beyond our trial, as long as the researcher performs the same processes with us. 

To repeat our outcome, three parts should be kept. Part 1 is to use fmriprep software. Part 2 is to select Schaefer template to extract the time series. Part 3 is to find the time series of the 60 regions in the Schaefer template.

All the information about these 60 regions could be found in Supporting Information (S1 Table) and our github repository, including the label, name, MNI space coordination of each region. Also, the Schaefer template and the predictive model were uploaded together.

Our github repository address https://github.com/Clancy-wu/TaiChi-CFS-2022.

Comment 3: With regard to the limitations of the study, given the amount of data collected, is it possible that even an equally-sized set of completely random data would yield a similar number of FCs? This is an issue in large-scale genomics studies.

Response: Thanks for your kind reminders. As you said, the large-scale network analysis is to find result within a large number of data, but we do not think this is a trick in mathematic. 

First, using FCs to make prediction has been reported in before. About 12 years ago, the author Dosenbach accomplished 91% accuracy to make prediction via functional connections between regions of brain networks, and published article in the top journal SCIENCE (Dosenbach, 2010, Science, doi:10.1126/science.1194144). This means the process ‘predict via FCs with brain network’ is a trustworthy method.

Second, many robust steps were performed to make results reliable in our trial. In pre-process, we used a high-repeatability software ‘fmriprep’ to ensure that the time series we got is reliable. In statistical analysis, we used the strict multiple-correction method Bonferroni to ensure the result reliable. A random data cannot pass the multiple correction in such a strict level (α = 0.05/7 = 0.007 and α = 0.05/14 = 0.0036).

Third, our results fitted previous literature. Our outcomes focus on two networks, the DMN and left FPN. As we discussed in article, the DMN network was the most reported network in CFS patients in previous (three studies reported among seven studies in CFS area). And the FPN, especially the left, was found to work in physical exercise (including Tai Chi practice) via a meta-analysis (Yu Qian, Brain structure & function, 2021). Therefore, our result is an extension of the previous finding, not an irrelevantly random innovation with other studies.

To sum up, we think our finds within the amount of data in the article is much meaningful, and these cannot be replaced by the equally-sized random data.

Comment 4: Line 53: This statement about the relationship of Tai Chi to the broader area of Qigong is not necessarily universally held. It may be better to say that in some lineages of Tai Chi and with some teachers, there can be substantial overlap with practices found in Qigong. I note that in many cases, taijiquan is taught with a completely different emphasis and with completely different defining characteristics, especially if the goal is martial usage of jin and qi.

Response: Thanks for your kind reminders. We changed our description and re-wrote this paragraph [Line 52-64, Pg 3].

Comment 5: Line 60: Again, it is quite debatable whether Tai Chi is a representative Qigong. This also ignores the issue of whether the state defined simplified 24-style is an adequate representation of taijiquan in general. And, obviously, the teacher has the primary effect in terms of whether a student can actually learn the skills required for qigong or taijiquan.

Response: Thanks for your kind reminders. We changed our description and re-wrote this paragraph [Line 52-64, Pg 3].

Comment 6: Line 77: Change to "could be a mechanism".

Response: Thanks for your kind reminders. We realized that we are not rigorous enough in word, your carefulness helps us a lot. We revised this paragraph.

Comment 7: Line 85: This absolute statement about "traditional statistical [sic] methods" is quite overblown. In fact, many statistical techniques make relatively few assumptions or relatively mild assumptions. It can just as easily be said that most machine learning techniques suffer from a complete lack of theoretical underpinning, interpretability, or generalizability.

Response: Thanks for your kind reminders. Sorry for that we exaggerated the criticism for traditional statistical method to present the machine learning. We changed our description [Line 95-97, Pg 5]. 

Comment 8: Line 98: What would be the mechanism of action for improving the special functional connectivity?

Response: Thanks for your kind reminders. Sorry for that our wrong expression misled you. We revised our description [Line 107-108, Pg 5]:

Comment 9: Line 130: The phrase "was unclear" implies partial knowledge even though full knowledge was expected. Do the authors mean "blinded" here?

Response: Thanks for your kind reminders. Sorry for that our wrong expression misled you. We used ‘blinded’ to replace ‘unclear’ [Line 142, Pg 7].

Comment 10: Line 204: Why was the max-min normalization performed? It seems unnecessary given the statistical methods used.

Response: Thanks for your kind reminders. The SF-36 questionnaire contains eight parts — Physical Functioning, Role-Physical, Bodily Pain, General Health, Vitality, Social Functioning, Role-Emotional, Mental Health — and each part calculates differently with others.

For instance, the score of Physical Functioning equals (real score -10) / 20 * 100; the score of Role-Physical equals (real score - 4) / 4 * 100, etc.

The ‘max-min normalization’ represents the calculation of the score of each part via their own formula. The ‘average score ’ represents summing up scores of eight parts and then dividing eight (Table 1) [Line 242, Pg 11]. 

Therefore, we could use one value to show the average health statement of each subject, and to analyze the improvement of Tai Chi for the health statement (displayed in Table 1) via covariance analysis.

Comment 11: Line 313: "[M]achine learning is better at prediction than statistical inference" is likely a truism because machine learning pays nearly no attention to the issues of statistical inference.

Response: Thanks for your kind reminders. Sorry for our poor expression. We revised this sentence ‘machine learning only plays role in prediction rather than statistical inference’ [Line 383-384, Pg 18].

Comment 12: Line 347: Does the phrase "at the bottom of my life" correctly interpret the author's intent?

Response: Thanks for your kind reminders. Sorry for that my poor expression worried you. I revised this sentence ‘that they gave me much care in my trough time’ [Line 419, Pg 20]. Thanks a lot.

Comment 13: Figure 2: Change "Liner" to "Linear". Also, clarify that feature construction was performed only on training data.

Response: Thanks for your kind reminders. We revised Fig 2. 

Response to Review 2

Comment 1: Line 102: There is no power analysis, to define sample size. I would like to see a power analysis to be sure whether the sample size is enough or not for the fMRI analysis. Why do you choose 20 patients and 20 healthy control?

Response: Thanks for your kind reminders. Speaking honestly, we haven’t provided power analysis, because our sample size was defined by the median of previous literature. Despite the small sample size, we want to make explanations for our trial in two parts, why we choose 20:20 samples and whether this sample size could be enough to show our result.

Question 1: Why we choose 20:20 samples?

(1) In fMRI area, the cost of fMRI data acquisition for one subject is much expensive than other clinical trials. Shortly ago, Scott Marek published an article in Nature, saying that the sample with a good replication rate in fMRI analysis requires over thousands (Marek, S., Nature, 2022), which caused quite a bit of controversy. Of course, the larger the sample size, the more convincing the results are. However, to accomplish such a great sample size is very difficult.

In year 2017, Russell A. Poldrack obtained 548 studies in fMRI area at year 2011-2015, found that the median group size was 19 subjects in single group (Poldrack, R. A., Nature reviews. Neuroscience, 2017). In year 2020, Russell A Poldrack again pointed out, result from literature review showed that more than half of the samples comprised fewer than 50 people, in the machine learning & fMRI area (Poldrack, R. A., JAMA psychiatry, 2020). From those we can see, our trial samples size 20:20 was at the median level compared with the previous studies.

(2) Patients with CFS disease are hard to recruit. In the area of CFS and machine learning as well as fMRI, there were three previous studies (Provenzano et al., 2020; Sevel et al., 2018; Provenzano et al., 2020), two enrolled 38 CFS patients (both belongs to the author Provenzano), and one enrolled 18 CFS patients (author Sevel). Compared with them, our study enrolled 20 CFS patients, ranks the second. This also means that our trial was at the median level.

Taking those two parts into consideration, we defined the sample size of the trial into 20:20, and we think, under the median level sample size, our trial result could discover some things reliable, like the previous literature.

Question 2: Whether this sample size could be enough to show our result?

Under this median level sample size, we made efforts to increase the reliable and repeatability of our trial result in two parts.

(1) We are a longitudinal trial in CFS & machine learning & fMRI area. 

Scott Marek — the author we mentioned in Question 1 — highly recommends the intervention trial design with fMRI, and importantly, he doesn’t deny the value of the longitudinal design in the small-sample neuroimaging. He said in his Nature article “Within-person designs (for example, longitudinal) … or both (for example, interventions) frequently have increased measurement reliability and effect sizes … Thus, small-sample neuroimaging will always be critical for studying the human brain”. In fact, the previous three studies in CFS & machine learning & fMRI area (Provenzano et al., 2020; Sevel et al., 2018; Provenzano et al., 2020) were all were cross-sectional designs. And our trial could be the first longitudinal design in this study area. Thus, compared to them, whether in the size of dataset or the difficulty to accomplish the trial, our experiment is more valuable, as well as the result more meaningful.

(2) We performed a series of robust technology to ensure the repeatability of the results.

We applied the unified fmriprep workflow to process the dataset, the five-fold cross-validation to evaluate model performance, the strict multiple-correction Bonferroni method, and 10 repetitions to compute each result. All of these increased the repeatability of our trial as much as possible.

In summary, under the median level sample size and the robust calculation processes, we believe our results of the trial are reliable and worthy to be published. 

Comment 2: Line 138: MRI recording parameters have been given really weak. The structural (T1w) and functional recording parameters should be more clear and more details in terms of repeatability.

Response: Thanks for your kind reminders. We filled up the details of MRI recording parameters [Line 152-158, Pg 7]:

‘the resting-state echo-planar imaging sequence acquisition (time of repetition = 2,000 ms, time of echo = 30 ms, flip angle = 90°, phase encoding direction = A >> P, coverage = whole brain including cerebellum, field of view = 240 mm × 240 mm, matrix = 64 × 64, slice thickness = 3.5 mm, volumes = 240), the three-dimensional structure imaging adopting T1W1 sequence (time of repetition = 1900 ms, time of echo = 2.53 ms, coverage = whole brain including cerebellum, field of view = 250 mm × 250 mm, matrix = 256 × 256, slice thickness = 1.0 mm, volumes = 176)’.

Comment 3: ICA-AROMA tool is only capable of denoise movement artifacts. But the other artifacts such as respiratory, cardiac, or MRI device-based noise remain intact in resting-state data. Did the authors remove artifacts also manually? If they did not it should be written as a limitation.

Response: Thanks for your kind reminders. We add this into our limitation [Line 402-403, Page 19]:

‘Forth, although the denoise movement artifacts were removed, the other artifacts such as respiratory, cardiac and MRI device-based noise might remain influences to our data analysis.’

Comment 4: Authors use linear SVC for feature ranking, and then the most informative features are used to train a Random Forest classifier. The authors must justify why they did not perform classification with SVC. Besides, the feature ranking by SVC is based on having maximum accuracy by the SVC classifier. Can we guarantee that the same features will give the maximum classification result by the Random Forest? Since they already started with SVC we would like to see the classification results with SVC.

Response: Thanks for your kind reminders. We used linear SVC for features selection (decreased 160004 features into 60 features), and Random Forest classifier for prediction and features importance ranking (figure out which feature is important under 60 features). The workflow can be found in Fig 2.

Since our design is a longitudinal trial, not a cross-section, we would pay more attention to find that which brain areas work to the intervention, rather than to seek for highest accuracy. Generally speaking, the more features remained, the higher accuracy model would get, especially in cross-section trial. On the contrary, the target of our trial is to reduce the features number while maintaining a good accuracy as much as possible. This is the reason why we should perform ‘feature selection’ process. 

There are many methods for feature selection. In our trial, we chose a method named ‘Select From Model’ (referred to scikit-learn software document), the Model can apply tree-based model or linear model. Here, we used linear SVC model for feature selection. 

However, when the model is used for feature selection, it cannot be applied for classification in the meantime. The causing reason is that if the feature number is greatly reduced, the accuracy of the model will definitely decrease. For instance, in our trial, on the 160004 features, the linear SVC model we trained reached near 90% accuracy, but on the 60 features, it sharply reduced to half, because the model has been suited to the structure of 160004 features. 

Consequently, we used two model, the linear SVC is used for features selection (in train data) and the random forest is used for classification (in train data and test data). The linear SVC tell us which features related to the intervention and the random forest verified prediction ability of these features.

We choose to use linear SVC because this model often has a good performance in machine learning with wide application. We choose to use random forest because this model always has a good performance in neuro-image area. 

Comment 5: It would be better if we can see a more detailed comparative analysis of feature ranking and classification processes. For example (Line 224), they selected 60 of more than 160000 features. How the threshold value of 60 was determined? It would be great if we can see a plot of a number of most informative features vs obtained classification accuracy.

Response: Thanks for your kind reminders. We added the default threshold of features selection [Line 196, Pg 9], ‘…threshold = 1e-5…’.

From the document of scikit-learn software description, the default threshold of ‘Select From Model’ is 1e-5. We controlled feature selection via adjusting L1 value, thus forgot to explain the default threshold. Now we added.

As was said in Comment 4, the process of feature selection (a number of most informative feature) cannot apply the same model with the process of prediction (obtained classification accuracy). Feature selection (‘Select From Model’ method) is not a continues process that you can observe the feature number reduction step by step. Once we defined the L1 value, several features will be left, the L1 value size is irrelevant with the number of features left. We only can control L1 value from small to big, but the number of features will up and down without regular. Thus, the X-axis is hard to plot.

The obtained classification accuracy is generated by random forest model. When we got the selected features from the linear SVC (the most informative features), its initial performance was not good under the default random forest model, 50% - 60% actually. Then, we should tune the hyper-parameters of the random forest to increase accuracy. This largely depended on the researcher experience to know whether it really reach the best accuracy. You may cannot promise the best accuracy on each time of the most informative features you get.

Therefore, the plot is hard to make. The result we found in our trial was tried for thousands calculation. As we mentioned in the limitation, we may not have found all the important features, we just found a good unit of features and they predicted greater than before.

Comment 6: Line 186: Since the dataset is small, the cross-validation is wise to apply, but can we guarantee that 10 repeated runs would be theoretically enough to see the actual performance? A mathematical justification of this value “10” would be required. A plot of average accuracy vs. the number of repetitions would be helpful as well.

Response: Thanks for your kind reminders. We used five-fold cross-validation and repeated five-fold cross-validation on 10 times. In our trial, we performed five-fold cross-validation on training data for 10 repeats, we predicted on test data for 10 repeats, and applied permutation test on features ranking for 10 repeats. Actually, number 10 is not a value verified by mathematic, it is just generated by the conventional thinking. But for more credibility, we used permutation test to support the prediction accuracy score. We believe the score of permutation test and score of prediction on 10 repeats can complement each other to illustrate the performance of the random forest model, rather than relying on a certain score. 

If only one score should be chosen, the score of permutation test will be more reliable.

We added the result of the prediction accuracy per time on S4 Table, and ROC score of the random forest model on S4 Fig.

Response to Review 3

Comment 1: The main hypotheses of this paper are very general. The introduction cites some studies about the effects of Tai Chi in general, but the authors just say that machine learning will find some “special pattern” of FC differences in CFS, and that the regions/networks/connections showing differences at baseline will show some kind of non-specific changes after the training. While the authors discuss some of the particular changes in the context of the literature in their Discussion, the Introduction would be improved by at least including some more specific background information about which functional networks may be affected in CFS, and why Tai Chi may change those networks.

Response: Thanks for your kind reminders. We added more details in the introduction and revised the paragraph [Line 81-89, Pg 4]. 

Comment 2: In the “Clinical design and evaluation” section, the authors state that a clinical evaluator was “unclear about the grouping of subjects in the study.” Does this mean they were blinded? Or something else? If there was no waitlist or other control group and everyone received the same Tai Chi intervention, what were the evaluators unclear about?

Response: Thanks for your kind reminders. We revised this sentence [Line 141-144, Pg 7]:

‘All subjects underwent clinical evaluation and resting-state fMRI scanning from the scale evaluator, who was blinded about the grouping of subjects in the study. The decision that the subject who entered the CFS group was made by clinician through consultation’.

In our trial, the clinical doctor decided the group of each subject, and another person (a postgraduate in our team) was responsible for scale evaluation. The evaluator may guess the subject group, but he would not know the answer. So, we think the evaluator could be said ‘blinded’.

Comment 3: The authors state that “subjects were required to practice Tai Chi for 30 minutes per day by themselves at home.” Do the authors have data to report about adherence to this requirement (i.e., how much home practice was completed by each participant)? It then says “the Tai Chi teaching and family Tai Chi exercise of each subject were recorded, including live recording…” Does this mean that each home practice session was recorded in some way as well, or just the group or instructor-led sessions? Please clarify.

Response: Thanks for your kind reminders. We revised this sentence [Line 137-140, Page 6]:

‘During our trial, for each subject, the Tai Chi teaching was recorded by live recording, and the family Tai Chi exercise was recorded through video feedback and telephone follow-up, in order to supervise the quality of the practice. In the end, all subjects had completed the required exercise time’

Comment 4: The data acquisition parameters require more details and clarification. No anatomical scan is mentioned, only stating the “parameters of this machine scan.” The parameters reported appear to be for a functional EPI sequence, but this is not specified. Also, while the time of repetition is stated as 2mm, the total number of volumes (and the total duration of the scan) is not reported.

Response: Thanks for your kind reminders. We filled up the details of MRI recording parameters [Line 152-158, Pg 7]:

‘The fMRI data were acquired by a Siemens 3-T MRI scanner (Germany), and the parameters of this machine scan were as follows: the resting-state echo-planar imaging sequence acquisition (time of repetition = 2,000 ms, time of echo = 30 ms, flip angle = 90°, phase encoding direction = A >> P, coverage = whole brain including cerebellum, field of view = 240 mm × 240 mm, matrix = 64 × 64, slice thickness = 3.5 mm, volumes = 240), the three-dimensional structure imaging adopting T1W1 sequence (time of repetition = 1900 ms, time of echo = 2.53 ms, coverage = whole brain including cerebellum, field of view = 250 mm × 250 mm, matrix = 256 × 256, slice thickness = 1.0 mm, volumes = 176)’. 

Comment 5: The data preprocessing section states that smoothing was done at 2mm FWHM. This is much lower than us conventionally used. Typical guidelines suggest to use a FWHM equivalent to 2 or 3 times the size of the voxels. With a voxel size of 3.75x3.75x5mm as stated in the data acquisition section, we would expect a spatial smoothing kernel of at least 7-8mm FWHM. This step can have a significant impact on functional connectivity findings. Can the authors explain this unconventional preprocessing choice?

Response: Thanks for your kind reminders. We feel much sorry for this description mistake. The actual steps are that, voxel size normalized to 2 mm, and the FWHM for volume is [6 6 6]. This sentence has been revised [Line 167-168, Line 171, Pg 8]:

‘… voxel size normalizing to 2 mm… smoothing with Gaussian kernel of full width at half maximum of 6 mm’.

The 2 mm volume size and [6 6 6] of FWHM are the default setting in DPABISurf software and are recommended by Professor Chaogan Yan, who is the author of DPABISurf software.

Comment 6: In the Discussion, the words “decrease” and “lower” are used interchangeably, leading to some confusion as to whether it is referring to the assessment of baseline differences between CFS and healthy volunteers or to the changes resulting from the intervention. For instance, the sentence “Our trial found a similar decrease in the DMN” is confusing. The “trial” should refer to the Tai Chi intervention, but earlier the authors state that increased DMN FC was found following the intervention. So when discussing these results, the authors should clarify the distinction between the baseline differences and the changes observed following the trial/intervention.

Response: Thanks for your kind reminders. We adjusted our word description, using ‘lower’ on baseline and ‘decrease’ on the effect of the intervention. 

Comment 7: The discussion could be improved with more interpretation of the results related to the DMN (i.e. the post-intervention increase in DMN connectivity). The authors currently say that DMN is important and “plays a central role in healthy people and patients with various diseases”, but do not elaborate much. They mention a previous hypothesis of DMN underconnectivity in CFS patients, which suggests that DMN underconnectivity is associated with greater energy needs and reduced energy available to the individual. They say that Tai Chi could “improve the abnormal pattern of the DMN,” but they do not elaborate on this point. This seems like potentially the most important part of the discussion, so it would be good to see this result discussed more. They go on to highlight that increased inter-network DMN connectivity was found after the intervention, and state that this is “interesting”. This could be improved by a couple sentences/references explaining why this might be interesting and worthy of further investigation. The discussion of FPN-DMN connectivity that follows is great!

Response: Thanks for your kind reminders. We added more details about DMN and FPN-DMN in the Discussion [Line 337-353, Pg 16-17].

Comment 8: The axis labels for Figure 6A and 6B say “differences” but this could be clarified by stating whether this is post-pre or pre-post. What are the units of these differences?

Response: Thanks for your kind reminders. We revised figure 6 and added a striking label ‘Post- minus Pre-’. Besides, we revised the caption of figure 6 [Line 306-307, Page 15]:

However, the unit cannot be clarified, because they haven’t been defined any unit in the previous literature, especially in the value of the functional connection.

Thanks very much for your attention and time. Looking forward to hearing from you.

Sincerely,

Yours

Kang Wu

Dongzhimen Hospital, Beijing University of Chinese Medicine

Beijing, China

---

## [Decision Letter · Decision Letter 1]

9 Sep 2022

PONE-D-22-14151R1Tai Chi Increases Functional Connectivity and Decreases Chronic Fatigue Syndrome: A Pilot Intervention Study with Machine Learning and fMRI AnalysisPLOS ONE

Dear Dr. Li,

Thank you for submitting your manuscript to PLOS ONE. After careful consideration, we feel that it has merit but does not fully meet PLOS ONE’s publication criteria as it currently stands. Therefore, we invite you to submit a revised version of the manuscript that addresses the points raised during the review process.

We look forward to receiving your revised manuscript.

Kind regards,

Burak Yulug

Academic Editor

PLOS ONE

Reviewers' comments:

Reviewer's Responses to Questions

**Comments to the Author**

1. If the authors have adequately addressed your comments raised in a previous round of review and you feel that this manuscript is now acceptable for publication, you may indicate that here to bypass the “Comments to the Author” section, enter your conflict of interest statement in the “Confidential to Editor” section, and submit your "Accept" recommendation.

Reviewer #2: All comments have been addressed

Reviewer #3: (No Response)

2. Is the manuscript technically sound, and do the data support the conclusions?

Reviewer #2: Yes

Reviewer #3: Partly

3. Has the statistical analysis been performed appropriately and rigorously? 

Reviewer #2: Yes

Reviewer #3: Yes

4. Have the authors made all data underlying the findings in their manuscript fully available?

Reviewer #2: (No Response)

Reviewer #3: (No Response)

5. Is the manuscript presented in an intelligible fashion and written in standard English?

Reviewer #2: Yes

Reviewer #3: No

6. Review Comments to the Author

Reviewer #2: Many thanks for inviting me to review this paper.

Corrections seem to be enough after the major revision decision. This paper will help increase our understanding in functional neuroimagin analysis. I think the manuscript is suitable to publish in PLOS ONE.

Reviewer #3: Comment 1: The authors have added a paragraph to help set up some of their hypotheses about how Tai Chi is expected to change functional connectivity patterns in CFS patients. However, I do still have some concerns about the relevance and clarity of this paragraph.

Even if few intervention studies have been done on CFS with neural outcomes, are there any studies showing what the baseline neural correlates of CFS are? What are the mechanisms by which Tai Chi would act on these neural signatures of CFS?

The following is not a complete sentence: “While others studies indicated that Tai Chi could improve sleep disorders through increasing functional connections of the sub-regions of DMN, including the medial prefrontal cortex and the medial temporal lobe [38,39].”

Comment 2: A few things about this section on Clinical Design and Evaluation are still unclear to me. I read the translated version of the supplement, but the details there are also lacking. One question raised by the supplement was the mention of the McGill Pain Questionnaire and the Hamilton Depression scale as secondary outcome measures. Why were these not mentioned in the main manuscript (unless they were used in a different manuscript)?

Regarding this sentence: “All coaches in the trial were graduates with sports majors and many years of experience with Tai Chi, and they were required to understand the design of the trial and possess basic knowledge of CFS disease.” Were the Tai Chi coaches informed about which group each participant was assigned to (i.e. did they know which participants had CFS and which were healthy volunteers?)? Even if the teaching curriculum was identical for CFS vs HC subjects, coach/trainer awareness of group assignment can bias the teaching and therefore the results. It is also unclear whether CFS and HC participants participated in the same classes together, or whether they were taught separately (i.e. one class contained only CFS and another contained only HC subjects), or whether it was all individual instruction (no groups). Could the authors clarify these questions?

This sentence is still not clear to me: “The decision that the subject who entered the CFS group was made by clinician through consultation’. Is this meant to say something such as “A consulting clinician made each decision about whether each subject would be assigned to the CFS or HC group.”?

Regarding this sentence: “The clinical evaluation both in two groups was conducted using three scale questionnaires: the Fatigue Scale-14 (FS-14) for fatigue symptom assessment (the higher its score, the more serious is the fatigue); Pittsburgh Sleep Quality Index (PSQI) for sleep quality measurement around one month (the higher its score, the lower is the sleep quality); and the MOS 36-item short-form health survey (SF-36) for people’s healthy state evaluation (the higher its score, the healthier is the body)”. Table 1 in the Results shows the baseline values of these scales for each group, but no direct comparison is made. Did the two groups have significant differences in FS-14, PSQI, and SF-36 scores at baseline? What were the cutoffs or thresholds used by the evaluator to determine whether each participant should be assigned to CFS or HC? The table does show p-values for the post-pre differences on each measure within each of the two groups, but did the authors examine whether there was an interaction effect between group and timepoint (i.e. did the CFS group show larger changes on any of these measures than the HC group)?

Comment 3: Regarding this added sentence: “In the end, all subjects had completed the required exercise time.” If you have data on practice time completed, perhaps you could add that to a table or describe it in the text? If you don’t have specific numbers (minutes, hours, etc) for practice time, which would be preferable but not required, how was it determined that subjects did indeed complete the requirements? Was it verified via the video recordings of all practice sessions or by self-report from participants?

Comment 4: Thank you for adding the details, very helpful!

Comment 5: Great, thanks for the clarification about your smoothing process.

Comment 6: The changes look good.

Comment 8: The figure is a bit clearer now. However, one question remains about the significance of the results. You have mentioned in other parts of the paper that strict Bonferroni correction was applied. In the case of the partial correlation presented in Figure 6a, the p-value is 0.28. Even if that correlation and the one shown in 6b were the only two partial correlation tests you ran (were there others?), the corrected Bonferroni p-value for two tests would be 0.025, making the first one non-significant. Unless I am missing something, what is making you confident that this is a truly significant result?

Additional comments:

In some newly added lines in the Discussion, it says “we found that the increases of functional connections between the DMN and other networks may also correlate to the improvements of fatigue and sleep disorder (Table 1).” Table 1 does not show associations between DMN functional connectivity and fatigue/sleep disorder symptoms, so this reference might need to be updated?

On Line 392, you say that your results “proves the effectiveness of Tai Chi exercise for CFS alleviation.” “Proves” is a very strong word and I think much more work is needed before we have “proof” that Tai Chi is effective for CFS! I would use more cautiously optimistic language here. Even from a non-fMRI data perspective, you have not shown an interaction effect demonstrating that the symptom improvements after Tai Chi for the CFS group were significantly greater than those seen in the HC group. Unless you can show that, I would not use this kind of language in the paper, and even then I would be more cautious.

7. PLOS authors have the option to publish the peer review history of their article (what does this mean?). If published, this will include your full peer review and any attached files.

Reviewer #2: **Yes: **

Reviewer #3: No

---

## [Author Response · Author response to Decision Letter 1]

16 Sep 2022

Dear Editor,

We appreciate you and all the reviewers for your precious time in reviewing our paper and providing valuable comments. It was your valuable and insightful comments that led to possible improvements in the current version. The authors have carefully considered the comments and tried our best to address every one of them, and a final revised version was both submitted. The authors welcome further constructive comments if any.

Besides, please update our financial disclosure with ‘The funders had no role in study design, data collection and analysis, decision to publish, or preparation of the manuscript’.

Sincerely,

Yours

Kang Wu

Dongzhimen Hospital, Beijing University of Chinese Medicine

Beijing, China

Response to Review 3

Comment 1: The authors have added a paragraph to help set up some of their hypotheses about how Tai Chi is expected to change functional connectivity patterns in CFS patients. However, I do still have some concerns about the relevance and clarity of this paragraph.

Even if few intervention studies have been done on CFS with neural outcomes, are there any studies showing what the baseline neural correlates of CFS are? What are the mechanisms by which Tai Chi would act on these neural signatures of CFS?

The following is not a complete sentence: “While others studies indicated that Tai Chi could improve sleep disorders through increasing functional connections of the sub-regions of DMN, including the medial prefrontal cortex and the medial temporal lobe [38,39].”

Response: Thanks for your kind reminders.

The neural baselines of CFS were descripted before the paragraph you mentioned, including follows [Paragraph 3, Line 73-77]:

(a) “Charles [30] detected the decreased intrinsic connectivity within the left frontoparietal network (LFPN) and the decreased extrinsic connectivity involving the sensory motor network (SMN) and the salience network (SN) on CFS patients, both of which were related to the level of their fatigue symptom”, the “decreased intrinsic connectivity” and “decreased extrinsic connectivity” indicated the neural baselines of CFS.

(b) “Besides, the default mode network (DMN), which is focused on recently, showed connection disruptions in CFS [29,31] and other fatigue related disease [32]”, the “connection disruptions” indicated the neural baseline of CFS.

The effects of Tai Chi on brain networks and brain areas could be found as follows [Line 83-88]:

“A study illustrated the effect of Tai Chi in regulating the rest-state functional connections of the DMN and LFPN to enhance cognitive function on healthy people [37]” and “others studies indicated that Tai Chi could improve sleep disorders through increasing functional connections of the sub-regions of DMN, including the medial prefrontal cortex and the medial temporal lobe [38,39]”, the “DMN” and “LFPN” are the brain networks that Tai Chi acts with, and the “prefrontal cortex” and “medial temporal lobe” are the specific brain areas that Tai Chi works with.

Combining the above, we now add a sentence to clarify our purpose “It seemed that functions of Tai Chi and disfunctions of CFS were overlapped partly in brain networks, for instance, the DMN and LFPN [Line 87-88]”.

In this article, we hypothesis that the reorganization of brain network’s functional connectivity maybe the mechanism of Tai Chi treating CFS. Thus, which brain networks were participated in this reorganization is the study target of our research majorly. However, the mechanisms of why Tai Chi could change these brain networks and how Tai Chi made such a changed pattern were beyond this article.

Besides, thanks for your carefulness. We apologize for this sentence mistake. The sentence was revised as follows:

“Further, others studies indicated that Tai Chi could improve sleep disorders through increasing functional connections of the sub-regions of DMN, including the medial prefrontal cortex and the medial temporal lobe [38,39]”

Comment 2: A few things about this section on Clinical Design and Evaluation are still unclear to me. I read the translated version of the supplement, but the details there are also lacking. One question raised by the supplement was the mention of the McGill Pain Questionnaire and the Hamilton Depression scale as secondary outcome measures. Why were these not mentioned in the main manuscript (unless they were used in a different manuscript)?

Regarding this sentence: “All coaches in the trial were graduates with sports majors and many years of experience with Tai Chi, and they were required to understand the design of the trial and possess basic knowledge of CFS disease.” Were the Tai Chi coaches informed about which group each participant was assigned to (i.e. did they know which participants had CFS and which were healthy volunteers?)? Even if the teaching curriculum was identical for CFS vs HC subjects, coach/trainer awareness of group assignment can bias the teaching and therefore the results. It is also unclear whether CFS and HC participants participated in the same classes together, or whether they were taught separately (i.e. one class contained only CFS and another contained only HC subjects), or whether it was all individual instruction (no groups). Could the authors clarify these questions?

This sentence is still not clear to me: “The decision that the subject who entered the CFS group was made by clinician through consultation’. Is this meant to say something such as “A consulting clinician made each decision about whether each subject would be assigned to the CFS or HC group.”?

Regarding this sentence: “The clinical evaluation both in two groups was conducted using three scale questionnaires: the Fatigue Scale-14 (FS-14) for fatigue symptom assessment(the higher its score, the more serious is the fatigue); Pittsburgh Sleep Quality Index (PSQI)for sleep quality measurement around one month (the higher its score, the lower is the sleep quality); and the MOS 36-item short-form health survey (SF-36) for people’s healthy state evaluation (the higher its score, the healthier is the body)”. Table 1 in the Results shows the baseline values of these scales for each group, but no direct comparison is made. Did the two groups have significant differences in FS-14, PSQI, and SF-36 scores at baseline? What were the cutoffs or thresholds used by the evaluator to determine whether each participant should be assigned to CFS or HC? The table does show p-values for the post-pre differences on each measure within each of the two groups, but did the authors examine whether there was an interaction effect between group and timepoint (i.e. did the CFS group show larger changes on any of these measures than the HC group)?

Response: Thanks for your kind reminders. 

(1) In the protocol of our research, our team have collected five questionnaires (MPQ, PSQI, HAMD, SF-36, FS-14). However, only three scales were mentioned (PSQI, SF-36, FS-14) in this article. The reasons included: 

(a) As a part of our team, we proposed this imaged analysis roadmap (machine learning & fMRI & Tai Chi) to apply for data from the team leader. The other part of data was in another submitted article by my partner. 

(b) In this article, we majorly focus on the changes of neuro-image, especially in large-scale brain network. Thus, clinical indicators were only used to express that Tai Chi could improve CFS symptoms rather than to explore all the possible aspects that Tai Chi may affect for CFS. As was descripted in the head of our article, “Chronic fatigue syndrome (CFS)…is characterized by severe fatigue…sleep disturbance…and self-reported impairments in concentration as well as short-term memory [1]”, we think the changes of the three questionnaires we put in our article (including PSQI, SF-36, FS-14) were sufficient to proof the effectiveness of Tai Chi for CFS. For instance, the FS-14 was corresponded to the severe fatigue, the PSQI was corresponded to the sleep disturbance and the SF-36 was corresponded to the self-reported health statement.

(2) Thanks so much, we revised the sentence “All coaches …and they were required to know the design of the trial and the basic knowledge of CFS disease before they took part in this research. On each teaching classes, the subjects in the CFS group or HC group were mixed to be taught but coaches never knew about group information” [Line 137-139]. 

In our trial, we think that coaches are required to understand our research are their rights to be informed, which cannot be extended to misunderstand that coaches would know much details about processes of our research. Specifically speaking, at the beginning of our trial, we would tell the coach candidates what type of this study and how they would cooperate with us to complete the study. Once they understand what we would do and agree with our study, they would be invited to join us. After that, the only thing that coaches should do is to focus on teaching, and they would never know the group information within their classes. 

Besides, HC people and CFS patients were mixed to be taught in each class. In our research, when we recruited sufficient subjects to this trial, we would open a class to teach them for one month. So, HC and CFS were mixed.

(3) Thanks so much. We revised this sentence “A clinician made each decision about whether subject would be assigned to the CFS or HC group” [Line 146-147]. 

(4) Thanks so much. CFS is a kind of disease that diagnosis majorly depends on patient’s self-report. The including criteria could be found in [Line 115-118] “They had chronic fatigue …and it generally persists or relapses for more than six months…”. Generally, when the person met with this disease description, he would go to our hospital and then be invited to join the trial after clinician evaluation. The clinician evaluation is also majorly relied on asking.

(a) We haven’t made a baseline comparison because we think those two groups were totally different before the intervention. For instance, the CFS group absolutely would have higher scores of FS-14 than HC group, otherwise he would not be recruited into CFS group. Now, for more rigorous in our result, we add the baseline comparison in Table 1 and the analysis description sentence “…and Student’s t-test for independent samples was performed for BMI comparison, questionnaires baseline comparison and questionnaires post-intervention comparison” [Line 224-226].

(b) As we said, CFS is majorly depended on the patients’ self-report, and there still haven’t a clinical indicator to be the diagnostic criteria. Thus, the cutoffs and thresholds would also cannot be captured. The subject would be assigned to CFS or HC majorly depended on his self-report that whether he had a long-tern fatigue syndrome and met with our including criteria. 

(c) In this article, detecting the brain networks changes is our main purpose in this article. Thus, we think the t-test’s result would be enough to be the proof of the effectiveness of Tai Chi. Thanks for your kind advise, we now add the repeated measures analysis of variance in clinical measurements in Table 2 and results expression in [Line 248-260]. This helps us a lot, much thanks.

Comment 3: Regarding this added sentence: “In the end, all subjects had completed the required exercise time.” If you have data on practice time completed, perhaps you could add that to a table or describe it in the text? If you don’t have specific numbers (minutes, hours, etc) for practice time, which would be preferable but not required, how was it determined that subjects did indeed complete the requirements? Was it verified via the video recordings of all practice sessions or by self-report from participants?

Response: Thanks for your kind reminders. We revised this sentence “In the end, all subjects had completed the required exercise times and exercise frequency” to clarify our purpose. 

On the process of our trial, each subject was asked to complete the required exercise times (1 hour of teaching classes and 30 minutes of family classes) and exercise frequency (eight times for teaching classes and 20 times for family classes, totally 28 times exercise) [Line 134-135 for teaching classes, 139-140 for family classes]. For teaching classes, we supervised on site and recorded the exercise via live recording. For family classes, we supervised by their video feedback and telephone follow-up [Line 141-142]. Hence, we were confident to say “In the end, all subjects had completed…”. If you asked how we can make sure the truth of each video feedback, we may say that we tried our best to recognize. Anyway, we do have completed the record of each exercise, and we sincerely thought that this sentence desired to be kept, for the sake of our great efforts.

Comment 4: Thank you for adding the details, very helpful!

Response: Thank you.

Comment 5: Great, thanks for the clarification about your smoothing process.

Response: Thank you.

Comment 6: The changes look good.

Response: Thank you.

Comment 8: The figure is a bit clearer now. However, one question remains about the significance of the results. You have mentioned in other parts of the paper that strict Bonferroni correction was applied. In the case of the partial correlation presented in Figure6a, the p-value is 0.28. Even if that correlation and the one shown in 6b were the only two partial correlation tests you ran (were there others?), the corrected Bonferroni p-value for two tests would be 0.025, making the first one non-significant. Unless I am missing something, what is making you confident that this is a truly significant result?

Response: Thanks for your kind reminders. 

Well, actually, these two (Figure 6A and Figure 6B) are not under the same hypothesis. We put these two together just for convenience to display. However, we now realize that this action could confuse the readers. Sorry for that.

Partial covariance analysis was the further exploration for our main results. We may first wonder whether there was correlation between DMN and LFPN, and we got the answer (Figure 6B). Then, we detected the correlations between networks (DMN / LPFN) and clinical measurements (FS-14 / PSQI / SF-36), and we got the answer (Figure 6A). We thought these two results were under the different hypotheses, so we did not perform the Bonferroni correction.

However, not only did you propose the potential false-positive in Figure 6A, but we also found the inexplicable part between Figure 6A and our others results. To consider the data’s integrity and result’s consistency, we haven’t deleted Figure 6A in the previous manuscript. Thanks for your carefulness, we choose to delete the Figure 6A and all the explanation sentences about Figure 6A in Discussion.

Additional comments:

In some newly added lines in the Discussion, it says “we found that the increases of functional connections between the DMN and other networks may also correlate to the improvements of fatigue and sleep disorder (Table 1).” Table 1 does not show associations between DMN functional connectivity and fatigue/sleep disorder symptoms, so this reference might need to be updated?

Response: Thanks for your kind reminders. We revised this sentence “Differently, we found that the increases of functional connections between the DMN and other networks could be the reasons of improvements of fatigue and sleep disorder (Tables 1-4)” [Line 365-367].

In our trial, the FCs of DMN and other networks increased and clinical measurements improved, and we thought these could be the mechanism of Tai Chi treating CFS. However, previous literature only focused on the sub-regions within DMN and the improvements of clinical symptoms. That was why we used “Differently” in here. Thanks for your reminders, we have realized that it was better to use the “reason” rather than the “correlate” to express our meaning accurately. Hence, we revised our description. Thanks again, your kind advise helped me a lot.

On Line 392, you say that your results “proves the effectiveness of Tai Chi exercise for CFS alleviation.” “Proves” is a very strong word and I think much more work is needed before we have “proof” that Tai Chi is effective for CFS! I would use more cautiously optimistic language here. Even from a non-fMRI data perspective, you have not shown an interaction effect demonstrating that the symptom improvements after Tai Chi for the CFS group were significantly greater than those seen in the HC group. Unless you can show that, I would not use this kind of language in the paper, and even then I would be more cautious.

Response: Thanks for your kind reminders. We changed this word with “illustrates” [Line 407]. Thanks again, your carefulness helped me a lot.

---

## [Decision Letter · Decision Letter 2]

16 Nov 2022

Tai Chi Increases Functional Connectivity and Decreases Chronic Fatigue Syndrome: A Pilot Intervention Study with Machine Learning and fMRI Analysis

PONE-D-22-14151R2

Dear Dr. Li,

We’re pleased to inform you that your manuscript has been judged scientifically suitable for publication and will be formally accepted for publication once it meets all outstanding technical requirements.

Kind regards,

Burak Yulug

Academic Editor

PLOS ONE

<quillbot-extension-portal></quillbot-extension-portal>

---

## [Editor Report · Acceptance letter]

22 Nov 2022

PONE-D-22-14151R2 

Tai Chi Increases Functional Connectivity and Decreases Chronic Fatigue Syndrome: A Pilot Intervention Study with Machine Learning and fMRI Analysis 

Dear Dr. Li:

I'm pleased to inform you that your manuscript has been deemed suitable for publication in PLOS ONE. Congratulations! Your manuscript is now with our production department. 

Kind regards, 

on behalf of

Dr. Burak Yulug 

Academic Editor

PLOS ONE